# LANGUAGE MODEL DECODING AS DIRECT METRICS OPTIMIZATION

**Haozhe Ji**    **Pei Ke**[*]    **Hongning Wang**    **Minlie Huang**[*]
The CoAI Group, DCST, BNRist, Tsinghua University, Beijing 100084, China
jihaozhe@gmail.com   aihuang@tsinghua.edu.cn

## ABSTRACT

Despite the remarkable advances in language modeling, current mainstream decoding methods still struggle to generate texts that align with human texts across different aspects. In particular, sampling-based methods produce less-repetitive texts which are often disjunctive in discourse, while search-based methods maintain topic coherence at the cost of increased repetition. Overall, these methods fall short in achieving holistic alignment across a broad range of aspects. In this work, we frame decoding from a language model as an optimization problem with the goal of strictly matching the expected performance with human texts measured by multiple metrics of desired aspects simultaneously. The resulting decoding distribution enjoys an analytical solution that scales the input language model distribution via a sequence-level energy function defined by these metrics. And most importantly, we prove that this induced distribution is guaranteed to improve the perplexity on human texts, which suggests a better approximation to the underlying distribution of human texts. To facilitate tractable sampling from this globally normalized distribution, we adopt the Sampling-Importance-Resampling technique. Experiments on various domains and model scales demonstrate the superiority of our method in metrics alignment with human texts and human evaluation over strong baselines.

## 1 INTRODUCTION

Although pre-trained on large corpora of human texts with scaled up sizes, existing autoregressive language models (LMs) (Radford et al., 2019; Brown et al., 2020; Zhang et al., 2022) are still struggling to produce human-like texts measured in various aspects, such as repetition, coherence, and consistency (Pillutla et al., 2021; Dou et al., 2022). Existing decoding methods are mainly driven to address two main mis-specifications of an LM's distribution: (i) The long tail of the distribution is *unreliable* (Holtzman et al., 2020), such that sampling from these low-probability regions often produces low-quality contents that are incoherent. (ii) The mode of the distribution is *degenerated* (Welleck et al., 2020), where samples

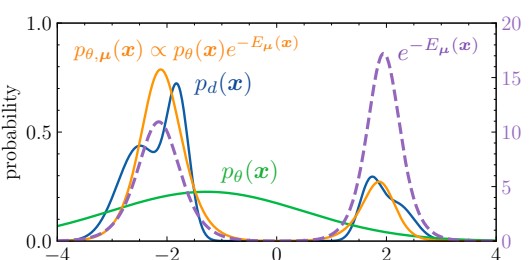

Figure 1: The decoding distribution $p_{\theta,\boldsymbol{\mu}}$ induced by DAEMON scales the input LM distribution $p_\theta$ with a sequence-level energy function $E_{\boldsymbol{\mu}}$, which leads to a more accurate recovery of the underlying data distribution $p_d$.

with high probabilities exhibit low diversity with repetitive patterns. As a result, sampling-based decoding methods (Fan et al., 2018; Holtzman et al., 2020; Meister et al., 2022) use various truncation strategies to avoid sampling from the unreliable long tail of the distribution, while recent search-based methods (Li et al., 2022; Su et al., 2022) incorporate additional contrastive objectives to avoid the collapse of degenerated repetitions. Since these two mis-specifications reside at opposing extremes of the probability spectrum, current decoding methods inevitably concentrate on

---

[*]Corresponding Author.

just one of them which addresses only a limited subset of aspects. Although heuristic designs and sophisticated hyper-parameter tuning allow trade-offs, these approaches usually cannot effectively align with human texts with respect to a broad range of critical aspects simultaneously.

Attempts have been made to fix the mis-specification issue of LM distribution by directly augmenting the standard Maximum Likelihood Estimation (MLE) with auxiliary training objectives (Welleck et al., 2020; Su et al., 2022; Xu et al., 2022). However, exposure bias (Chiang & Chen, 2021; Arora et al., 2022) prevents the effectiveness of such attempts. Specifically, since during training the autoregressive LM is conditioned on the ground-truth context, it is not guaranteed that the imposed properties in these training objectives would be preserved during decoding time, where the context is progressively generated by the LM itself. On the other hand, approaches based on Reinforcement Learning (RL) (Ranzato et al., 2016; Yu et al., 2017) address the exposure bias issue, but often encounter challenges to maintain proximity to the distribution of human texts (characterized by a low perplexity) (Caccia et al., 2020). Overall, these methods do not guarantee a general enhancement over the standard training paradigm, owing to the potential conflicts between their designated objectives and MLE (Lin et al., 2021b). More related work discussion is provided in Appendix B.

In this work, we focus on the decoding route and present a novel framework, **D**ecoding **A**s Dir**E**ct **M**etrics **O**ptimizatio**N** (DAEMON) that explicitly targets at aligning desired aspects with human texts. DAEMON frames decoding from a language model as an optimization problem with the goal of locating the optimal decoding distribution where sampled texts can strictly match with human texts in multiple evaluation metrics simultaneously. Formally, given the input LM distribution $p_\theta$ learned on the human text distribution $p_d$, DAEMON searches for the decoding distribution $q$ that minimizes the *reverse* Kullback-Leibler divergence (KL), $D_{\text{KL}}(q\|p_\theta)$, subject to the constraints of matching the expected evaluation metric scores under $q$ and $p_d$. We choose the reverse KL to induce the decoding distribution $q$, as it forces $q$ to recover the major probability masses within the support of $p_\theta$ (Huszar, 2015; Malinin & Gales, 2019), which contains mostly high-quality samples. Moreover, besides directly enforcing alignment on chosen metrics, we also rigorously prove that the optimization problem guarantees an improvement of the solution over the input LM in perplexity, which indicates a more general gain in aligning with human texts.

In addition to the theoretical guarantee, the decoding distribution induced by DAEMON also enjoys an analytical solution denoted as $p_{\theta,\boldsymbol{\mu}}$. It scales the *locally normalized* LM distribution $p_\theta$ with a sequence-level energy function $E_{\boldsymbol{\mu}}$ which depicts the underlying distribution $p_d$ from various perspectives by satisfying the corresponding constraints. In Figure 1, we visualize $p_{\theta,\boldsymbol{\mu}}$ in an illustrative example where the energy captures the disjoint regions of modes in $p_d$, which empowers the input LM distribution $p_\theta$ to facilitate a better approximation of $p_d$. To enable tractable sampling from $p_{\theta,\boldsymbol{\mu}}$, which is globally normalized over the space of all possible sequences, we adopt the Sampling-Importance-Resampling (SIR) technique (DB, 1988; Smith & Gelfand, 1992) that first samples candidates from $p_\theta$ and then resamples based on the importance weight defined by the energy function. We empirically demonstrate the effectiveness of DAEMON in open-ended text generation by considering a wide range of critical aspects including repetition, coherence, diversity, and information content across different model scales and data domains. Experimental results show that DAEMON outperforms strong decoding baselines in both automatic evaluation of metrics alignment with human texts and human evaluation.

## 2 METHOD: DECODING AS DIRECT METRICS OPTIMIZATION

We consider conditional language generation from a pre-trained language model specified by the distribution $p_\theta$, where the model is provided with a relatively short prefix $\boldsymbol{x}_{\leq t_0} = \{x_t\}_{t=1}^{t_0}$ of length $t_0$ and required to generate a continuation that results in a full text $\hat{\boldsymbol{x}}_{\leq T} = \{\hat{x}_t\}_{t=1}^{T}$ of total length $T$. In the following, the subscript of $\boldsymbol{x}_{\leq T}$ is omitted for convenience. Instead of directly sampling from $p_\theta$, we look for a decoding distribution induced from $p_\theta$ to produce human-like texts measured by a set of chosen metrics. For example, in the canonical top-$k$ sampling (Fan et al., 2018), the decoding distribution is obtained by truncating the conditional distribution $p_\theta(x_t|\boldsymbol{x}_{1:t-1})$ to keep the top-$k$ candidates at every decoding step, so as to improve the reliability of generated content.

Ideally, a perfect decoding distribution $q_{\text{opt}}$ assigns an arbitrary text sample $\boldsymbol{x}$ with the probability equals to $p_d(\boldsymbol{x})$, where $p_d$ is the underlying distribution of human texts. In practice, this is infeasible since we only have samples from $p_d$, rather than $p_d$ itself. However, given a text evaluation metric

we are interested in (such as repetition and coherence), formally $f : \mathcal{X} \to \mathbb{R}$ that maps $\boldsymbol{x}$ in the text space $\mathcal{X}$ to a real value, an alternative criterion for measuring the closeness from $q_{\mathrm{opt}}$ to $p_d$ can be achieved by matching the expectation of $f$ under $q_{\mathrm{opt}}$ and $p_d$, i.e., $\left| \mathbb{E}_{\hat{\boldsymbol{x}} \sim q_{\mathrm{opt}}}[f(\hat{\boldsymbol{x}})] - \mathbb{E}_{\boldsymbol{x} \sim p_d}[f(\boldsymbol{x})] \right|$. This expectation-matching criterion, commonly employed in prior studies (Holtzman et al., 2020; Meister et al., 2022; Su et al., 2022) as an empirical evaluation of the resemblance of generated texts against human texts. This forms the basis of our proposed optimization-based decoding framework that directly aligns the generated texts with human texts against the set of chosen text evaluation metrics.

## 2.1 FORMULATION OF THE OPTIMIZATION PROBLEM

At the core of our proposed decoding framework, we look for the optimal solution $q_{\mathrm{opt}}$ of the following constrained optimization problem which searches for the decoding distribution $q$ closest to the given LM distribution $p_\theta$ and strictly matching the expectations on the generated texts with that of human texts measured by a set of chosen evaluation metrics:

$$q_{\mathrm{opt}} = \arg\min_{q \in \mathcal{P}} D_{\mathrm{KL}}(q \| p_\theta) \tag{1}$$

$$s.t.\ \mathbb{E}_{\hat{\boldsymbol{x}} \sim q}[f_k(\hat{\boldsymbol{x}})] = \mathbb{E}_{\boldsymbol{x} \sim p_d}[f_k(\boldsymbol{x})], \quad k \in \{1, \cdots, K\},$$

where $\boldsymbol{f} = \{f_k\}_{k=1}^{K}$ is a set of evaluation metrics we concern, and $\mathcal{P}$ is the set of all probability densities in the input space $\mathcal{X}$.

The formulation of our proposed optimization problem hinges on our key insight of constructing a decoding distribution from a language model to acquire samples that closely resemble human texts. The constraints, as defined to match the performance of evaluated metrics on generations with those obtained on human texts, explicitly ensure this goal in expectation. The *reverse* KL divergence in the optimization objective, i.e., $D_{\mathrm{KL}}(q \| p_\theta)$, restricts the decoding distribution $q$ to deviate minimally from the LM distribution $p_\theta$ by encouraging **mode-seeking** behavior, which satisfies the quality-demanding nature of decoding. Although the forward KL is extensively employed as an optimization objective in learning data-driven probabilistic models (Radford et al., 2019), its induced distribution is shown to mismatch with the quality assessment of human (Pang & He, 2021) by overestimating the long tail of the target distribution (Ji et al., 2023) due to its **mean-seeking** behavior. More discussion is provided in Appendix C. We believe the learning and decoding phases posit different goals: the former is to capture all modes in the data, while the latter is to decode high-quality ones. Hence, we require the decoding distribution to only explore within the support of the given LM distribution, which is naturally realized by minimizing the reverse KL. Existing truncation-based sampling (Fan et al., 2018; Welleck et al., 2020; Meister et al., 2022) can be deemed as a heuristic that shares the same spirit of ours in maintaining a finite reverse KL, since the support of the truncated distribution is always the strict subset of the support of the given LM distribution.

The formulation of our optimization problem is also known as information projection in previous literature of information geometry (Csiszár & Matús, 2000; Nielsen, 2020), which can be deemed as finding the projection of $p_\theta$ on the manifold of distributions that constrains $p_d$. In the following proposition, we show that it actually leads to a nice analytical solution. The full proof of Proposition 1 is provided in Appendix A.1.

**Proposition 1.** *The distribution that solves the optimization problem (1) is in the form of:*

$$p_{\theta, \boldsymbol{\mu}}(\boldsymbol{x}) \propto p_\theta(\boldsymbol{x}) \exp\left[ -E_{\boldsymbol{\mu}}(\boldsymbol{x}) \right], \quad \forall \boldsymbol{x} \in S(p_{\theta, \boldsymbol{\mu}}) \tag{2}$$

*where $E_{\boldsymbol{\mu}}(\boldsymbol{x}) = \boldsymbol{\mu}^\top \boldsymbol{f}(\boldsymbol{x})$ and $S(p) = \{\boldsymbol{x} : p(\boldsymbol{x}) > 0\}$ is the support of distribution $p$. $\boldsymbol{\mu} \in \mathbb{R}^K$ is determined by the constraints in (1).*

The unnormalized form of $p_{\theta, \boldsymbol{\mu}}(\boldsymbol{x})$, also known as the Energy-Based Model (EBM) (Rosenfeld et al., 2001; Hinton, 2002; LeCun et al., 2006), takes advantage from both the given LM distribution $p_\theta$ and the energy function $E_{\boldsymbol{\mu}}(\boldsymbol{x})$ that serves as a sequence-level assessment about the satisfaction of constraints measured by the evaluation metrics. The contribution of individual metrics to the overall alignment performance is characterized by the derived coefficients $\boldsymbol{\mu} = \{\mu_k\}_{k=1}^{K}$. Decoding from Eq. (2) requires determining $\boldsymbol{\mu}$ and tractable sampling from the normalized density, which will be discussed in §2.3. In the next subsection, we take a step further and demonstrate that the optimal solution of the problem (1) guarantees a theoretical improvement in perplexity of human texts.

## 2.2 THEORETICAL IMPROVEMENT IN PERPLEXITY

Although explicitly driving the generation to align with human texts under the chosen evaluation metrics is appealing, we are still confronted with the question of whether the resulting decoding distribution is generally a better approximation to the underlying distribution of human texts. For most existing heuristic decoding methods, a distribution-level evaluation (e.g., perplexity) is infeasible because of their ad-hoc treatments on the input LM distribution. For example, distribution truncation (Fan et al., 2018; Welleck et al., 2020; Meister et al., 2022) leads to a sparse support which is smaller than the underlying distribution of human texts, while heuristic searching algorithms (Li et al., 2022; Su et al., 2022) such as beam search do not have a parametric decoding distribution. Martins et al. (2020) proposed a variant of the standard perplexity, $\epsilon$-perplexity by smoothing a sparse distribution, which still can not faithfully reflect the true perplexity of the truncated distribution.

For the decoding distribution derived from the proposed optimization problem, we show that not only is the perplexity feasible to compute, but it also improves the perplexity of human texts against the original LM distribution. The full proof is provided in Appendix A.2.

**Proposition 2.** *The optimal solution* $q_{\text{opt}}$ *of the optimization problem (1) satisfies:*

1. $S(q_{\text{opt}}) \supseteq S(p_d)$, *where* $S(p) = \{\boldsymbol{x} : p(\boldsymbol{x}) > 0\}$.

2. $H(p_d, q_{\text{opt}}) = H(p_d, p_\theta) - D_{\text{KL}}(q_{\text{opt}} \| p_\theta)$, *where* $H(p, q) = -\sum_{\boldsymbol{x}} p(\boldsymbol{x}) \log q(\boldsymbol{x})$.

*Proof sketch.* The proof starts with the convexity of the set $\mathcal{C}$ of distributions that satisfy the constraints in Eq. (1). We then consider $p_\alpha = (1 - \alpha)q_{\text{opt}} + \alpha p_d \in \mathcal{C}$, for $\alpha \in [0, 1]$. The key insight is the following observation:

$$\frac{\partial}{\partial \alpha} D_{\text{KL}}(p_\alpha \| p_\theta) \Big|_{\alpha=0} = H(p_d, p_\theta) - H(p_d, q_{\text{opt}}) - D_{\text{KL}}(q_{\text{opt}} \| p_\theta). \tag{3}$$

$\partial D_{\text{KL}}(p_\alpha \| p_\theta) / \partial \alpha$ can also be written as the limit of $[D_{\text{KL}}(p_\alpha \| p_\theta) - D_{\text{KL}}(q_{\text{opt}} \| p_\theta)] / \alpha$ which is non-negative when $\alpha \to 0^+$ due to the optimality of $q_{\text{opt}}$. Therefore, for Eq. (3) to be non-negative, we must have $q_{\text{opt}}(\boldsymbol{x}) \neq 0$ for any $\boldsymbol{x} \in S(p_d)$ (otherwise it converges to $-\infty$), which proves the first claim. Next, given $S(q_{\text{opt}}) \supseteq S(p_d)$, there exists some $\alpha' < 0$ such that $p_{\alpha'}$ is a probability density function, which by definition also belongs to $\mathcal{C}$. Therefore, $[D_{\text{KL}}(p_{\alpha'} \| p_\theta) - D_{\text{KL}}(q_{\text{opt}} \| p_\theta)] / \alpha'$ is non-positive when $\alpha' \to 0^-$, leading to $\partial D_{\text{KL}}(p_\alpha \| p_\theta) / \partial \alpha |_{\alpha'=0} = 0$, which proves the second claim. $\square$

The first outcome of Proposition 2 establishes the feasibility of computing perplexity under $p_{\theta, \boldsymbol{\mu}}$ when evaluated using the underlying human text distribution $p_d$. And the second result reveals the perplexity improvement over $p_\theta$: $2^{H(p_d, q_{\text{opt}})} < 2^{H(p_d, p_\theta)}$, due to the non-negativity of $D_{\text{KL}}(q_{\text{opt}} \| p_\theta)$. Note that the perplexity of $q$ is defined as $2^{H(p_d, q)}$.

Intuitively, more powerful constraints in the optimization problem that better measure the alignment with human texts cause a larger deviation from the input LM distribution, which in turn leads to a better approximation of underlying human text distribution, and thus a lower perplexity.

## 2.3 DECODING FROM THE OPTIMAL SOLUTION

In this section, we describe the method to decode from the sampling distribution derived from the optimization problem (1). First, we describe our method to estimate the coefficients $\boldsymbol{\mu}$ by satisfying the constraints with a conditional proposal distribution. Then we introduce a tractable sampling method to obtain samples from the decoding distribution defined by the EBM.

### 2.3.1 COEFFICIENTS ESTIMATION

The only degrees of freedom in the analytical solution of the optimal decoding distribution $p_{\theta, \boldsymbol{\mu}}$ are the coefficients $\boldsymbol{\mu} = \{\mu_k\}_{k=1}^K$ in the energy function $E_{\boldsymbol{\mu}}(\boldsymbol{x})$, whose optimal values $\boldsymbol{\mu}_{\text{opt}}$ can be estimated by first calculating $\hat{\boldsymbol{F}} = \mathbb{E}_{\boldsymbol{x} \sim p_{\theta, \boldsymbol{\mu}}}[\boldsymbol{f}(\boldsymbol{x})]$ and then approximating the target expectation $\boldsymbol{F} = \mathbb{E}_{\boldsymbol{x} \sim p_d}[\boldsymbol{f}(\boldsymbol{x})]$ to satisfy the constraints with iterative gradient updates. Note that this procedure is done on a small development set once for all before the inference stage.

First, $\hat{\boldsymbol{F}}$ can be estimated by Weighted Importance Sampling (WIS) (Geweke, 1989; Hesterberg, 1995) which first obtains $N$ i.i.d. trajectories $\{\hat{\boldsymbol{x}}^i\}_{i=1}^N \sim p_\theta$, and then computes the weighted sum of $\boldsymbol{f}(\hat{\boldsymbol{x}}^i)$ with importance weight proportional to $\exp(-E_{\boldsymbol{\mu}}(\hat{\boldsymbol{x}}^i))$ normalized over all trajectories. As the asymptotic bias and variance of $\hat{\boldsymbol{F}}$ estimated by WIS are both proportional to $N^{-1}$ (Hesterberg, 1995), the target expectation can be approximated with required estimation error by drawing enough samples from the proposal. Detailed derivation of WIS is provided in Appendix A.4.1.

---

**Algorithm 1** $\boldsymbol{\mu}_{\text{opt}}$ estimation with WIS

---

**Input:** $p_\theta$, $\boldsymbol{F}$, learning rate $\alpha$
**Output:** $\boldsymbol{\mu}_{\text{opt}}$
1: Initialize $\boldsymbol{\mu}$ randomly
2: Sample trajectories $\{\hat{\boldsymbol{x}}^i\}_{i=1}^N \sim p_\theta$
3: **repeat**
4:     $\hat{\boldsymbol{F}} \leftarrow \frac{\sum_{i=1}^N \exp(-E_{\boldsymbol{\mu}}(\hat{\boldsymbol{x}}^i))\boldsymbol{f}(\hat{\boldsymbol{x}}^i)}{\sum_{i=1}^N \exp(-E_{\boldsymbol{\mu}}(\hat{\boldsymbol{x}}^i))}$
5:     $\boldsymbol{\mu} \leftarrow \boldsymbol{\mu} - \alpha\nabla_{\boldsymbol{\mu}}\sqrt{\frac{1}{K}\|1 - \hat{\boldsymbol{F}}/\boldsymbol{F}\|_2^2}$
6: **until** convergence
7: $\boldsymbol{\mu}_{\text{opt}} \leftarrow \boldsymbol{\mu}$

---

Next, given $\hat{\boldsymbol{F}}$ as a parametric function of the variable $\boldsymbol{\mu}$, we propose to approximate the target expectation $\boldsymbol{F}$ by minimizing the Root Mean Squared Relative Error (Shcherbakov et al., 2013), $\sqrt{\frac{1}{K}\|1 - \hat{\boldsymbol{F}}/\boldsymbol{F}\|_2^2}$ where the estimation error of each $f_k$ is normalized to the same scale. Then the optimal coefficient $\boldsymbol{\mu}_{\text{opt}}$ is obtained by iteratively updating $\boldsymbol{\mu}$ until convergence, i.e., reaching a desired error level. The algorithm of coefficients estimation is shown in Algorithm 1. We also analyze the convergence of $\boldsymbol{\mu}$ in Appendix G and find it insensitive to initialization. Runtime analysis of Algorithm 1 is provided in Appendix E, which demonstrates the advantage over the typical hyperparameter search procedure for most other decoding methods (Meister et al., 2022; Li et al., 2022).

### 2.3.2 CONDITIONAL SAMPLING FROM EBM

Sampling from the decoding distribution defined by EBM in Eq. (2) is non-trivial, given that it is globally normalized over the whole sequence space. We first present the conditional probability of sampling a continuation $\boldsymbol{x}_{>t_0}$ from $p_{\theta,\boldsymbol{\mu}}$ given a prefix $\boldsymbol{x}_{\leq t_0}$:

$$p_{\theta,\boldsymbol{\mu}}(\boldsymbol{x}_{>t_0}|\boldsymbol{x}_{\leq t_0}) = p_\theta(\boldsymbol{x}_{>t_0}|\boldsymbol{x}_{\leq t_0})\exp\Big[-E_{\boldsymbol{\mu}}(\boldsymbol{x}_{\leq t_0}, \boldsymbol{x}_{>t_0})\Big]/Z(\boldsymbol{x}_{\leq t_0}), \tag{4}$$

where $Z(\boldsymbol{x}_{\leq t_0}) = \mathbb{E}_{\boldsymbol{x}'_{>t_0} \sim p_\theta(\cdot|\boldsymbol{x}_{\leq t_0})}[\exp(-E_{\boldsymbol{\mu}}(\boldsymbol{x}_{\leq t_0}, \boldsymbol{x}'_{>t_0}))]$ is the marginalization over future tokens sampled from the conditional proposal given the prefix. The detailed derivation is provided in Appendix A.5.1. As direct sampling from this auto-regressive factorization is computationally prohibitive (Deng et al., 2020), we instead turn to a particle-based approximation of $p_{\theta,\boldsymbol{\mu}}$ using the sampling-importance-resampling (SIR) technique (DB, 1988; Smith & Gelfand, 1992).

Specifically, we first leverage the given LM $p_\theta$ as a proposal to generate a set of $M$ plausible continuation candidates $\{\hat{\boldsymbol{x}}_{>t_0}^i\}_{i=1}^M$ given the prefix $\boldsymbol{x}_{\leq t_0}$ in parallel. Then the final generation result is resampled from the distribution defined by the importance weight which is proportional to $\exp(-E_{\boldsymbol{\mu}}(\boldsymbol{x}_{\leq t_0}^i, \hat{\boldsymbol{x}}_{>t_0}^i))$ normalized over all candidates $\{\hat{\boldsymbol{x}}^i\}_{i=1}^M$. We present the SIR approximation procedure of the conditional probability $\hat{p}_{\theta,\boldsymbol{\mu}}^M(\cdot|\boldsymbol{x}_{\leq t_0})$ in Appendix A.5.2. In the limit of $M \rightarrow$

---

**Algorithm 2** Conditional Sampling with SIR

---

**Input:** $p_\theta$, $E_{\boldsymbol{\mu}}$, prefix $\boldsymbol{x}_{\leq t_0}$, $M$, $\tau$
**Output:** continuation $\boldsymbol{x}_{>t_0}$
1: **for** $i \leftarrow 1$ to $M$ **do**        ▷ In parallel
2:     Sample $\hat{\boldsymbol{x}}_{>t_0}^i \sim p_\theta^\tau(\cdot|\boldsymbol{x}_{\leq t_0})$
3:     Compute $w_i \leftarrow \exp(-E_{\boldsymbol{\mu}}(\boldsymbol{x}_{\leq t_0}, \hat{\boldsymbol{x}}_{>t_0}^i))$
4: **end for**
5: Sample $j \sim \text{Categorical}\Big(\frac{w_1}{\sum_{i=1}^M w_i}, \cdots, \frac{w_M}{\sum_{i=1}^M w_i}\Big)$
6: Set $\boldsymbol{x}_{>t_0} \leftarrow \hat{\boldsymbol{x}}_{>t_0}^j$

---

$\infty$, the empirical distribution $\hat{p}_{\theta,\boldsymbol{\mu}}^M(\cdot|\boldsymbol{x}_{\leq t_0})$ induced by SIR recovers the exact conditional distribution $p_{\theta,\boldsymbol{\mu}}(\cdot|\boldsymbol{x}_{\leq t_0})$ for arbitrary $\boldsymbol{x}_{>t_0}$. Skare et al. (2003) proved that the point-wise relative error of the empirical distribution induced by SIR is $O(M^{-1})$ (see Theorem 2.1 in the original paper). In practice where $M$ is finite, we propose to sample from the the temperature modulated proposal $p_\theta^\tau$ with lower temperature $\tau$ to increase the chance of obtaining high-quality candidates within a realistic computational budget. The algorithm of conditional sampling is shown in Algorithm 2. In fact, various existing sampling methods can be used for candidate sampling, we choose to use temperature sampling as it preserves the feasibility to compute perplexity (see Appendix A.3). We provide complexity and runtime analysis of Algorithm 2 and baseline decoding methods in Appendix F.

## 3 EXPERIMENT

### 3.1 DATASETS

We evaluate our method on the Wikipedia and News domain for open-ended text generation. For the Wikipedia domain, the data comes from documents in the Wikitext-103 corpus (Merity et al., 2017). For the News domain, the data comes from news articles in Wikinews[1]. We follow the data pre-processing procedure suggested by Li et al. (2022), and randomly select 512 samples as the development set for hyper-parameter tuning for all decoding methods. The data statistics of each domain and detailed data pre-processing steps are provided in Appendix J.

### 3.2 EVALUATION METRIC SETTINGS

In this section, we introduce the set of evaluation metrics we consider in aligning with human texts, which correspond to $f$ in Eq. (1). These metrics cover a wide range of aspects including repetition, coherence, diversity, and information content.

**Repetition.** We evaluate repetition at both sequence level and token level. The sequence-level metric measures the portion of duplicate $n$-grams in the generated texts (Welleck et al., 2020): $\text{SEQ-REP-N} = 100 \times (1 - \frac{|\text{unique } n\text{-grams}(\hat{\boldsymbol{x}})|}{|\text{total } n\text{-grams}(\hat{\boldsymbol{x}})|})$ where $\hat{\boldsymbol{x}}$ is the generated text (SR-N in short). The token-level metric measures the average frequency of each generated token reoccuring in the previous $l$ tokens (Fu et al., 2021; Ji & Huang, 2021): $\text{TOK-REP-L} = 100 \times (\frac{1}{|\hat{\boldsymbol{x}}|} \sum_{t=1}^{|\hat{\boldsymbol{x}}|} \mathbb{1}[\hat{x}_t \in \hat{\boldsymbol{x}}_{t-l-1:t-1}])$ (TR-L in short). We adopt SR-N with $n = \{2, 3, 4\}$ and TR-L with $l = \{8, 16, 32\}$, respectively.

**Coherence.** We evaluate coherence following Su et al. (2022) by calculating the cosine similarity between the sentence embedding of the prefix $\boldsymbol{x}_{\leq t_0}$ and the generated continuation $\hat{\boldsymbol{x}}_{>t_0}$: $\text{COH} = 100 \times \cos(\text{emb}(\boldsymbol{x}_{\leq t_0}), \text{emb}(\hat{\boldsymbol{x}}_{>t_0}))$ where $\text{emb}(\cdot)$ is parametrized by the pre-trained sentence embedding model SimCSE (Gao et al., 2021) based on RoBERTa (Liu et al., 2019).

**Diversity.** We evaluate diversity following Li et al. (2022) by aggregating the $n$-gram repetition rate of $n = \{2, 3, 4\}$: $\text{DIV} = 100 \times \prod_{\text{N}=2}^{4}(1 - \text{SEQ-REP-N})$. DIV reflects the overall lexical diversity of the text at different levels of granularity.

**Information Content.**[2] We evaluate the average amount of information contained per word given the preceding contexts, by calculating the exponential of the entropy rate on the generated text $\hat{\boldsymbol{x}}$ using a language model: $e^{\text{ENT}} = \exp(-\frac{1}{|\hat{\boldsymbol{x}}|} \sum_{t=1}^{T} \log p_{\text{LM}}(\hat{x}_t|\hat{\boldsymbol{x}}_{<t}))$ (Shannon, 1951; Braverman et al., 2020). A low $e^{\text{ENT}}$ suggests that the information in the text is redundant, while a high $e^{\text{ENT}}$ indicates high surprisal in the text according to the language model. In our experiment, we use a general-domain language model GPT-2 XL to calculate the log probability of the generated texts.

Note that these metrics are also used in the automatic evaluation part of our experiment (§3.5) to measure how well the generated texts align with human texts in different aspects. Thus, the criterion for these evaluation metrics is the *closeness* between the metric scores of generated texts and those of human references.

In addition, we also report **MAUVE** score (Pillutla et al., 2021) (MAU in short) which measures the distributional similarity between the set of generated texts and that of references by calculating the area under the divergence frontier of the two empirical distributions. As this metric cannot provide an evaluation score that corresponds to a certain aspect for each text sample, we only adopt it to assess the final performance of different decoding methods. We use GPT-2 XL to extract features from texts which was restricted to a maximum length of 256.

### 3.3 BASELINES AND IMPLEMENTATION DETAILS

We thoroughly compare DAEMON with various sampling-based and search-based methods in particular. We consider three canonical sampling-based methods: **Top-k** sampling (Fan et al., 2018), **Nucleus** sampling (Holtzman et al., 2020) and **Typical** decoding (Meister et al., 2022). For search-

---

[1] http://www.wikinews.org.
[2] https://en.wikipedia.org/wiki/Information_content.

| Method | | Wikipedia | | | | | | News | | | | | |
|---|---|---|---|---|---|---|---|---|---|---|---|---|---|
| | | SR-4 | TR-32 | COH | DIV | $e^{\text{ENT}}$ | MAU | SR-4 | TR-32 | COH | DIV | $e^{\text{ENT}}$ | MAU |
| | Reference | 0.48 | 21.3 | 62.3 | 92.5 | 23.2 | - | 0.29 | 18.7 | 66.6 | 94.1 | 13.8 | - |
| GPT-2 XL | Greedy | 60.9 | 65.5 | 60.2 | 8.03 | 2.29 | 59.7 | 53.2 | 58.2 | 63.8 | 13.2 | 2.19 | 65.2 |
| | Top-k | 2.11 | 23.4 | 60.9 | 87.8 | 10.1 | 77.8 | 0.95 | 20.3 | 64.7 | 91.7 | 8.17 | 96.3 |
| | Nucleus | 1.19 | 20.0 | 57.3 | **92.4** | 17.3 | 78.3 | 0.80 | 18.7 | 60.8 | 93.5 | 11.0 | 95.3 |
| | Typical | 0.81 | 17.4 | 54.9 | 94.5 | 30.1 | 78.7 | 0.42 | 16.9 | 57.2 | 95.3 | 18.2 | 95.0 |
| | CD | 1.31 | 28.2 | 68.7 | 85.9 | 7.55 | 77.8 | 0.63 | 23.2 | 71.2 | 90.5 | 6.55 | 95.1 |
| | CS | 1.78 | 23.0 | 56.9 | 90.6 | 5.25 | 83.3 | 0.77 | 19.2 | 63.6 | **94.1** | 4.18 | 95.7 |
| | DAEMON | **0.42** | **22.5** | **62.5** | 92.2 | **22.8** | **88.1** | **0.18** | **18.7** | **66.3** | 94.5 | **13.7** | **97.4** |
| OPT-6.7B | Greedy | 54.8 | 60.4 | 62.0 | 0.12 | 2.78 | 64.8 | 45.2 | 51.0 | 63.6 | 0.22 | 2.72 | 70.7 |
| | Top-k | 2.44 | 24.1 | 61.3 | 86.6 | 13.9 | 77.5 | 1.53 | 19.9 | **65.7** | 90.5 | 10.7 | 95.7 |
| | Nucleus | 2.33 | 21.9 | 59.1 | 88.6 | 18.9 | 80.1 | 1.37 | 19.2 | 63.3 | 91.5 | 12.2 | 95.3 |
| | Typical | 1.06 | 19.6 | 57.0 | 92.9 | 31.9 | 77.7 | 0.95 | 17.7 | 59.4 | **93.7** | 19.4 | 95.2 |
| | CD | 2.90 | 26.5 | 68.6 | 82.3 | 11.7 | 78.6 | 1.93 | 21.0 | 71.7 | 87.5 | 9.20 | 95.2 |
| | CS | 1.13 | 21.7 | 57.7 | 91.8 | 8.72 | 83.3 | 1.18 | 18.3 | 62.9 | 93.2 | 6.69 | 94.0 |
| | DAEMON | **0.38** | **21.6** | **62.3** | **92.6** | **22.7** | **90.7** | **0.25** | **18.8** | 64.8 | 94.9 | **13.6** | **97.2** |

Table 1: Main results of automatic evaluation on the Wikipedia and News domain using GPT-2 XL and OPT-6.7B. For all metrics, the best scores are the *closest* to the human scores except for MAU, which is better when *higher*. The best score is in **boldface** and the second best is underlined.

based methods, besides vanilla **Greedy** decoding, we also consider two recent methods that maximize the contrastive objectives: Contrastive Decoding (**CD**) (Li et al., 2022) and Contrastive Search (**CS**) (Su et al., 2022).

To demonstrate the effectiveness of our method across different language model families and scales, we consider GPT-2 XL (1.5B) (Radford et al., 2019) and OPT-6.7B (Zhang et al., 2022) as the base models for all decoding methods. For baselines, we follow the hyper-parameter settings in the original papers which are shown to work well in general. For DAEMON in the main results, we use the nine metrics (described in §3.2) in the constraints. During sampling, we set the size of candidate set from the proposal model $M = 25$ as it balances efficiency and performance. We set $\tau = 0.97$ for the Wikipedia domain and $\tau = 0.99$ for the News domain. We leave more implementation details of the baselines and DAEMON in Appendix J.2.

### 3.4 HUMAN EVALUATION

We further conduct human evaluation to assess the quality of generated texts. We consider three widely used criteria for open-ended generation: *Fluency*, *Coherence*, and *Informativeness* (van der Lee et al., 2019). Specifically, *Fluency* is characterized by the grammatical correctness and naturalness of the text without repetition; *Coherence* is characterized by topic maintenance with the input prefix and well-structured discourse; *Informativeness* is characterized by the adequacy of elaborating engaging details in a coherent plot. We randomly sample 100 prefixes and conduct pair-wise comparisons between DAEMON and baselines. Three annotators on Amazon Mechanical Turk are hired to choose the better continuation (i.e., win, lose, or tie) from the ones generated by our method and the baselines in terms of the three criteria above. More detailed settings of human evaluation are provided in Appendix D.

### 3.5 MAIN RESULTS

**Automatic Evaluation.** We first present the main results obtained via automatic evaluation in Table 1, where we benchmark our method against strong baselines across different domains and model scales under the evaluation metrics described in §3.2. Due to the space limit, we only present representative metrics and leave the full results in Appendix J.3. DAEMON outperforms all baselines in aligning with human texts on four aspects including repetition, coherence, diversity, and information content, and it also achieves the highest MAUVE score across different domains (Wikipedia / News) and model scales (1.5B / 6.7B). The consistent performance improvement demonstrates the effectiveness and generalizability of DAEMON. Notably, DAEMON achieves the lowest sequence-level repetition (indicated by SR-4) across all settings and maintains human-level coherence and high MAUVE score at the meantime, compared with baselines that have a similar repetition level, e.g.,

| Model | Wikipedia | | News | |
| --- | --- | --- | --- | --- |
| | ori | imp | ori | imp |
| GPT-2 XL | 23.1 | **22.0** | 13.9 | **13.1** |
| OPT-6.7B | 16.4 | **16.2** | 10.8 | **10.2** |

Table 2: Perplexity evaluation results. "`ori`" is the original perplexity of the LM distribution. "`imp`" is the improved perplexity of the optimal decoding distribution.

| Ours vs. | Fluency | | Coherence | | Informativeness | |
| --- | --- | --- | --- | --- | --- | --- |
| | Win | Lose | Win | Lose | Win | Lose |
| CD | **0.54** | 0.35 | **0.48**$^*$ | 0.36 | **0.48**$^*$ | 0.27 |
| CS | **0.53**$^*$ | 0.34 | **0.47**$^*$ | 0.29 | **0.41** | 0.33 |
| Nucleus | **0.54**$^*$ | 0.33 | **0.66**$^*$ | 0.15 | **0.45**$^*$ | 0.30 |
| Typical | **0.53**$^*$ | 0.30 | **0.62**$^*$ | 0.19 | **0.44**$^*$ | 0.23 |

Table 3: Human evaluation results that compare our method with baselines on Wikipedia domain. $^*$ indicates a significance with p-value $< 0.005$.

the Typical decoding method. The results indicate that DAEMON is the only method that effectively aligns multiple aspects with human texts without hard trade-off among those aspects.

For sampling-based methods, Nucleus sampling generally preserves more diverse candidates than Top-k sampling and achieves better diversity at human-level at the cost of low coherence. Typical decoding over-emphasizes the long tail of the distribution, which produces texts with highly diverse lexicality (highest diversity) but severe topic shift (lowest coherence). On the other hand, search-based methods generally have substantially lower information score than sampling-based methods, which indicate the redundancy of information in the generated texts. Specifically, CD achieves the highest coherence score at the expense of having the worst diversity except for greedy decoding. By analyzing its generated texts, we found that CD tends to repeat tokens from the previous context (indicated by high TR-32), which could be a bias captured by SimCSE to compute representations with high similarity. Additionally, we found that the MAUVE score of CS is generally higher than CD when the evaluation length is set to 256, which contradicts the previous result (Li et al., 2022) where the length is truncated to 128. Our result is consistent with our observation that CD generates semantically more repetitive texts its longer generations. To demonstrate the universability of our method, we additionally conduct an experiment on a text summarization dataset in Appendix I.

**Perplexity Evaluation.** We calculate the perplexity of DAEMON's decoding distribution, which can be directly computed using a proposal model with temperature modulation. The derivation of perplexity calculation is presented in Appendix A.3. From the results in Table 2, we observe that the perplexity improvement over the original LM distribution is consistent across model sizes and data domains. Notably, the perplexity improvement is more obvious for the model with a smaller size (GPT-2 XL), which manifests the advantage of our approach in enhancing the capacity of the autoregressive LM in modeling language with a constrained computational budget.

**Human Evaluation.** We conduct pairwise human evaluation and selectively compare our method against strong baselines including Contrastive Decoding (CD), Contrastive Search (CS), Nucleus sampling, and Typical decoding. As shown in Table 3, DAEMON is more preferred than all four baselines in fluency, coherence, and informativeness respectively on the domain of Wikipedia according to the human judgment. Specifically, DAEMON generates significantly more coherent texts compared to all the baselines. We provide a list of qualitative cases produced by different decoding methods in Appendix J.4 to help readers comprehend the difference in their generation quality.

### 3.6 ABLATION STUDY

**Ablating Metrics in Constraints.** We ablate the metrics in the constraints of DAEMON to investigate their individual contributions to the overall alignment performance. In Table 4, we present the results of ablating different metrics while keeping other settings verbatim. The ablation experiments are obtained using GPT-2 XL on the Wikipedia domain. We first observe that for most metrics, removing them lead to drastic deterioration in the corresponding metric scores, e.g., SR-4 ($0.42 \rightarrow 3.57$), COH ($62.5 \rightarrow 57.6$), $e^{\text{ENT}}$ ($22.8 \rightarrow 19.9$). We also observe clear inter-dependence between certain metrics pairs, as removing one leads to notable performance drop in another, e.g., SR-N and DIV, $e^{\text{ENT}}$ and TR-L, which reflects the intrinsic correlation among the evaluated aspects.

**Number of Candidates for Resampling.** We then study the impact of the number of candidates for resampling ($M$ described in §2.3.2). In Figure 3, we present the results on five metrics and the relative decoding latency with a batch size of 1. We set the temperature of the proposal distribution to $1.0$ to isolate the impact of $M$. From the results, we observe that the alignment on all metrics generally improves with a larger $M$, which indicates better SIR approximation to the optimal decoding

| Metrics | SR-4 | TR-32 | COH | DIV | $e^{\text{ENT}}$ |
|---|---|---|---|---|---|
| Reference | 0.48 | 21.3 | 62.3 | 92.5 | 23.2 |
| DAEMON | 0.42 | 22.5 | 62.5 | 92.2 | 22.8 |
| w/o SR-N | **3.57** | 22.6 | 63.0 | **86.8** | 22.1 |
| w/o TR-L | 0.19 | 20.2 | 62.0 | 93.9 | 25.1 |
| w/o COH | 0.37 | 22.3 | **57.6** | 92.6 | 23.8 |
| w/o DIV | 0.30 | 22.1 | 62.4 | 93.0 | 23.3 |
| w/o $e^{\text{ENT}}$ | 0.55 | **23.0** | 63.1 | 91.5 | **19.9** |

Table 4: Ablation results of different metrics where rows in gray are results with corresponding metrics removed. "SR-N" and "TR-L" denote the set of metrics with $n = 2, 3, 4$ and $l = 8, 16, 32$ respectively.

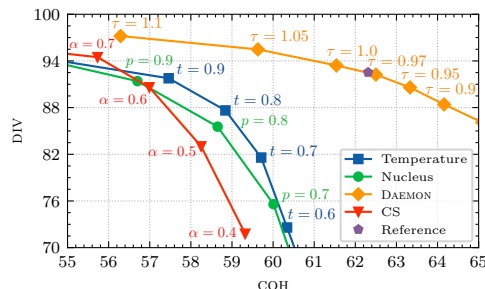

Figure 2: COH versus DIV when tuning the temperature $\tau$ of the proposal model of DAEMON and hyper-parameters of other baselines.

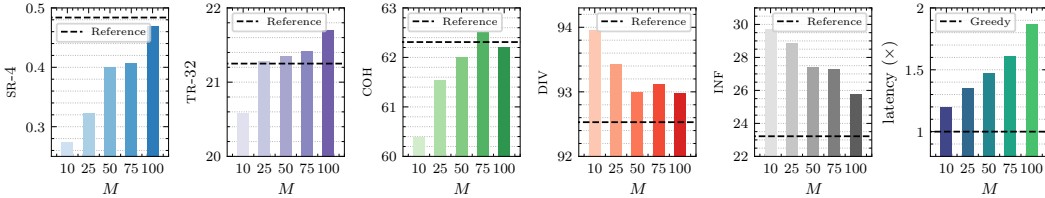

Figure 3: Ablation results of varying the number of candidates for resampling ($M$). Results on the five metrics are compared with the reference and the latency is relative to Greedy decoding.

distribution. Specifically, the convergence rate of different metric varies, e.g., DIV and $e^{\text{ENT}}$ converge slower than TR-32 and COH with the increase of $M$. Finally, increasing $M$ inevitably incurs higher decoding latency, and thus we choose $M = 25$ with slightly lower temperature in the main result to maintain both efficiency and performance. We test the robustness of our method by investigating the performance of different metrics when we optimize a single metric in Appendix H.

**Temperature When Sampling from the Proposal Model.** As suggested by (Caccia et al., 2020; Zhang et al., 2020), quality and diversity are two important aspects which can be traded off by sweeping hyper-parameters (e.g., temperature) to alter the sharpness of the distribution. For DAEMON, we tune the temperature of the proposal model ($\tau$ described in §2.3.2) with other settings unchanged and plot the curve in the dimensions of coherence and diversity in Figure 2. We also plot the result of tuning the hyper-parameters of different baseline methods. We first observe that DAEMON dominates the compared baselines in terms of coherence for all interested diversity level. Second, DAEMON is able to achieve human-level performance on these two aspects by slightly tuning the temperature lower, which demonstrates the effectiveness and practicality of our approach.

## 4  CONCLUSION AND FUTURE WORK

In this study, we introduce Decoding as Direct Metrics Optimization (DAEMON), a decoding framework that explicitly aligns the generations with human texts under various aspects, e.g., coherence, repetition, and etc. The induced sampling distribution harnesses candidates generated by an autoregressive LM, which are re-weighted according to a sequence-level energy function. We demonstrate both theoretical and empirical benefits of DAEMON, which outperforms strong decoding baselines in both human evaluation and automatic evaluation in terms of metrics alignment to human texts, perplexity improvement over the original LM, and superior quality-diversity trade-off.

As for the future work of DAEMON, we consider exploring directions to generalize the framework, e.g., extending the equality constraints to more general constraint types, such as inequalities and structural equations. It is also necessary to consider more aspects along with evaluation metrics beyond text quality, e.g., human value. This can therefore complement with other training-time alignment methods, such as RLHF. And finally more efficient method to sample from the distribution defined by the EBM is also important to ensure the practicality of DAEMON. Overall, we firmly believe this work paves the way for advanced methods that guide the language model towards desired behavior by incorporating constraints that capture intended regularities.

ACKNOWLEDGMENTS

This work was supported by the National Science Foundation for Distinguished Young Scholars (with No. 62125604), the NSFC projects (with No. 62306160 and No. 61936010), the China National Postdoctoral Program for Innovative Talents (No. BX20230194), and the China Postdoctoral Science Foundation (No. 2023M731952).

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

# A DERIVATIONS AND PROOFS

## A.1 PROOF OF PROPOSITION 1

**Proposition 1.** *The distribution that solves the optimization problem (1) is in the form of:*

$$p_{\theta,\boldsymbol{\mu}}(\boldsymbol{x}) \propto p_\theta(\boldsymbol{x}) \exp\Big[-E_{\boldsymbol{\mu}}(\boldsymbol{x})\Big], \quad \forall \boldsymbol{x} \in S(p_{\theta,\boldsymbol{\mu}}) \tag{2}$$

*where $E_{\boldsymbol{\mu}}(\boldsymbol{x}) = \boldsymbol{\mu}^\top \boldsymbol{f}(\boldsymbol{x})$ and $S(p) = \{\boldsymbol{x} : p(\boldsymbol{x}) > 0\}$ is the support of distribution $p$. $\boldsymbol{\mu} \in \mathbb{R}^K$ is determined by the constraints in (1).*

*Proof.* The optimization problem with full constraints is as follows:

$$\arg\min_q D_{\mathrm{KL}}(q \| p_\theta) \tag{5}$$

$$s.t. \quad \mathbb{E}_{\hat{\boldsymbol{x}} \sim q}[f_k(\hat{\boldsymbol{x}})] = \mathbb{E}_{\boldsymbol{x} \sim p_d}[f_k(\boldsymbol{x})], \quad k \in \{1, \cdots, K\} \tag{6}$$

$$q(\boldsymbol{x}) \geq 0, \quad \forall \boldsymbol{x} \in \mathcal{X} \tag{7}$$

$$\sum_{\boldsymbol{x} \in \mathcal{X}} q(\boldsymbol{x}) = 1, \tag{8}$$

where Eq. (7) and (8) are implicit constraints that guarantee $q$ to be a probability distribution over $\mathcal{X}$. The Lagrangian of the above optimization problem is:

$$L(q, \boldsymbol{\mu}, \boldsymbol{\lambda}, \tau) = \sum_{\boldsymbol{x} \in \mathcal{X}} q(\boldsymbol{x}) \log \frac{q(\boldsymbol{x})}{p_\theta(\boldsymbol{x})} + \boldsymbol{\mu}^\top \Big[\sum_{\boldsymbol{x} \in \mathcal{X}} q(\boldsymbol{x}) \boldsymbol{f}(\boldsymbol{x}) - \boldsymbol{F}\Big] - \sum_{\boldsymbol{x} \in \mathcal{X}} q(\boldsymbol{x})\lambda(\boldsymbol{x}) + \tau\Big[1 - \sum_{\boldsymbol{x} \in \mathcal{X}} q(\boldsymbol{x})\Big], \tag{9}$$

where $\boldsymbol{f}, \boldsymbol{\mu} \in \mathbb{R}^K, \boldsymbol{\lambda} \in \mathbb{R}^{|\mathcal{X}|}, \boldsymbol{F} = \mathbb{E}_{\boldsymbol{x} \sim p_d}[\boldsymbol{f}_k(\boldsymbol{x})]$.

The optimal set of solutions $(q_{\mathrm{opt}}, \boldsymbol{\mu}_{\mathrm{opt}}, \boldsymbol{\lambda}_{\mathrm{opt}}, \tau_{\mathrm{opt}})$ satisfy the Karush-Kuhn-Tucker conditions:

$$\log \frac{q_{\mathrm{opt}}(\boldsymbol{x})}{p_\theta(\boldsymbol{x})} + 1 + \boldsymbol{\mu}_{\mathrm{opt}}^\top \boldsymbol{f}(\boldsymbol{x}) - \lambda_{\mathrm{opt}}(\boldsymbol{x}) - \tau_{\mathrm{opt}} = 0, \quad \forall \boldsymbol{x} \in \mathcal{X} \tag{10}$$

$$\sum_{\boldsymbol{x} \in \mathcal{X}} q_{\mathrm{opt}}(\boldsymbol{x}) \boldsymbol{f}(\boldsymbol{x}) = \boldsymbol{F}, \quad \sum_{\boldsymbol{x} \in \mathcal{X}} q_{\mathrm{opt}}(\boldsymbol{x}) = 1, \quad q_{\mathrm{opt}}(\boldsymbol{x}) \geq 0, \quad \forall \boldsymbol{x} \in \mathcal{X} \tag{11}$$

$$\lambda_{\mathrm{opt}}(\boldsymbol{x}) \geq 0, \quad \lambda_{\mathrm{opt}}(\boldsymbol{x})q_{\mathrm{opt}}(\boldsymbol{x}) = 0, \quad \forall \boldsymbol{x} \in \mathcal{X} \tag{12}$$

We first obtain the form of $q_{\mathrm{opt}}$ from Eq. (10):

$$q_{\mathrm{opt}}(\boldsymbol{x}) = p_\theta(\boldsymbol{x}) \exp\Big[-\boldsymbol{\mu}_{\mathrm{opt}}^\top \boldsymbol{f}(\boldsymbol{x}) + \lambda_{\mathrm{opt}}(\boldsymbol{x})\Big]/Z, \tag{13}$$

where $Z = \sum_{\boldsymbol{x} \in \mathcal{X}} p_\theta(\boldsymbol{x}) \exp[-\boldsymbol{\mu}_{\mathrm{opt}}^\top \boldsymbol{f}(\boldsymbol{x}) + \lambda_{\mathrm{opt}}(\boldsymbol{x})]$ is the normalizing constant due to Eq. (11).

For every $\boldsymbol{x}'$ such that $q_{\mathrm{opt}}(\boldsymbol{x}') > 0$, from Eq. (12) we have $\lambda_{\mathrm{opt}}(\boldsymbol{x}') = 0$, which thereby proves Eq. (2).

$\square$

## A.2 PROOF OF PROPOSITION 2

**Proposition 2.** *The optimal solution $q_{\mathrm{opt}}$ of the optimization problem (1) satisfies:*

*1. $S(q_{\mathrm{opt}}) \supseteq S(p_d)$, where $S(p) = \{\boldsymbol{x} : p(\boldsymbol{x}) > 0\}$.*

*2. $H(p_d, q_{\mathrm{opt}}) = H(p_d, p_\theta) - D_{\mathrm{KL}}(q_{\mathrm{opt}} \| p_\theta)$, where $H(p, q) = -\sum_{\boldsymbol{x}} p(\boldsymbol{x}) \log q(\boldsymbol{x})$.*

*Proof.* Denote the set of probability densities that satisfy the constraints in the optimization problem (1) as $\mathcal{C}$:

$$\mathcal{C} = \{q \in \mathcal{P} : \mathbb{E}_{\boldsymbol{x} \sim q}[f_k(\boldsymbol{x})] = \mathbb{E}_{\boldsymbol{x} \sim p_d}[f_k(\boldsymbol{x})], \forall k \in [K]\}. \tag{14}$$

Then we consider $p_\alpha = (1 - \alpha)q_{\mathrm{opt}} + \alpha p_d \in \mathcal{C}$ for $\alpha \in [0, 1]$, where $q_{\mathrm{opt}}$ is the optimal distribution that solves the optimization problem. Due to the convexity of $\mathcal{C}$, $p_\alpha$ also belongs to $\mathcal{C}$.

Next we consider the following derivative $\partial D_{\mathrm{KL}}(p_\alpha \| p_\theta)/\partial\alpha$ at $\alpha = 0$:

$$\frac{\partial}{\partial\alpha} D_{\mathrm{KL}}(p_\alpha \| p_\theta)\Big|_{\alpha=0} = \frac{\partial}{\partial\alpha}\left(\sum_{\boldsymbol{x}} p_\alpha(\boldsymbol{x}) \log \frac{p_\alpha(\boldsymbol{x})}{p_\theta(\boldsymbol{x})}\right)\Bigg|_{\alpha=0} \tag{15}$$

$$= \left(\sum_{\boldsymbol{x}} \frac{\partial p_\alpha(\boldsymbol{x})}{\partial\alpha} \log \frac{p_\alpha(\boldsymbol{x})}{p_\theta(\boldsymbol{x})} + \frac{\partial p_\alpha(\boldsymbol{x})}{\partial\alpha}\right)\Bigg|_{\alpha=0} \tag{16}$$

$$= \left[\sum_{\boldsymbol{x}} \left(\log \frac{p_\alpha(\boldsymbol{x})}{p_\theta(\boldsymbol{x})} + 1\right)(p_d(\boldsymbol{x}) - q_{\mathrm{opt}}(\boldsymbol{x}))\right]\Bigg|_{\alpha=0} \tag{17}$$

$$= \left(\sum_{\boldsymbol{x}} (p_d(\boldsymbol{x}) - q_{\mathrm{opt}}(\boldsymbol{x})) \log \frac{p_\alpha(\boldsymbol{x})}{p_\theta(\boldsymbol{x})}\right)\Bigg|_{\alpha=0} \tag{18}$$

$$= \sum_{\boldsymbol{x}} (p_d(\boldsymbol{x}) - q_{\mathrm{opt}}(\boldsymbol{x})) \log \frac{q_{\mathrm{opt}}(\boldsymbol{x})}{p_\theta(\boldsymbol{x})} \tag{19}$$

$$= \sum_{\boldsymbol{x}} -p_d(\boldsymbol{x}) \log p_\theta(\boldsymbol{x}) + p_d(\boldsymbol{x}) \log q_{\mathrm{opt}}(\boldsymbol{x}) - q_{\mathrm{opt}}(\boldsymbol{x}) \log \frac{q_{\mathrm{opt}}(\boldsymbol{x})}{p_\theta(\boldsymbol{x})} \tag{20}$$

$$= H(p_d, p_\theta) - H(p_d, q_{\mathrm{opt}}) - D_{\mathrm{KL}}(q_{\mathrm{opt}} \| p_\theta) \tag{21}$$

While $\partial D_{\mathrm{KL}}(p_\alpha \| p_\theta)/\partial\alpha$ can also be written in the limit form:

$$\frac{\partial}{\partial\alpha} D_{\mathrm{KL}}(p_\alpha \| p_\theta)\Big|_{\alpha=0} = \lim_{\alpha\to0^+} \frac{D_{\mathrm{KL}}(p_\alpha \| p_\theta) - D_{\mathrm{KL}}(q_{\mathrm{opt}} \| p_\theta)}{\alpha} \tag{22}$$

Due to the optimality of $q_{\mathrm{opt}}$, $D_{\mathrm{KL}}(q_{\mathrm{opt}} \| p_\theta) \leq D_{\mathrm{KL}}(p_\alpha \| p_\theta)$, $\forall\alpha \in [0,1]$, which proves that $\partial D_{\mathrm{KL}}(p_\alpha \| p_\theta)/\partial\alpha|_{\alpha=0} \geq 0$. Therefore, for Eq. (19) to be non-negative, we must have $q_{\mathrm{opt}}(\boldsymbol{x}) \neq 0$ for any $\boldsymbol{x} \in S(p_d)$ (otherwise it converges to $-\infty$), which proves the first claim, $S(q_{\mathrm{opt}}) \supseteq S(p_d)$.

Next, we show that there exists some $\alpha' < 0$ such that $p_{\alpha'}$ is a probability density. For any $\boldsymbol{x} \in S(q_{\mathrm{opt}})$:

$$p_{\alpha'}(\boldsymbol{x}) = (1 - \alpha')q_{\mathrm{opt}}(\boldsymbol{x}) + \alpha' p_d(\boldsymbol{x})$$
$$= q_{\mathrm{opt}}(\boldsymbol{x}) + \alpha'[p_d(\boldsymbol{x}) - q_{\mathrm{opt}}(\boldsymbol{x})].$$

If $p_d(\boldsymbol{x}) - q_{\mathrm{opt}}(\boldsymbol{x}) = 0$, we always have $p_{\alpha'}(\boldsymbol{x}) = q_{\mathrm{opt}}(\boldsymbol{x}) \geq 0$.

If $p_d(\boldsymbol{x}) - q_{\mathrm{opt}}(\boldsymbol{x}) < 0$, we always have $p_{\alpha'}(\boldsymbol{x}) = q_{\mathrm{opt}}(\boldsymbol{x}) - \alpha'[q_{\mathrm{opt}}(\boldsymbol{x}) - p_d(\boldsymbol{x})] > 0$.

If $p_d(\boldsymbol{x}) - q_{\mathrm{opt}}(\boldsymbol{x}) > 0$, we set $\alpha' = -[\sup_{\boldsymbol{x}'} \frac{p_d(\boldsymbol{x}')}{q_{\mathrm{opt}}(\boldsymbol{x}')} - 1]^{-1}$ as $p_d(\boldsymbol{x})/q_{\mathrm{opt}}(\boldsymbol{x})$ is always greater than 1. Since $\alpha' \geq -\frac{q_{\mathrm{opt}}(\boldsymbol{x})}{p_d(\boldsymbol{x}) - q_{\mathrm{opt}}(\boldsymbol{x})}$, we have $p_{\alpha'}(\boldsymbol{x}) \geq 0$.

Therefore, we prove the non-negativity of $p_{\alpha'}$. Then it is trivial to show that $p_{\alpha'} \in \mathcal{C}$ by the linearity of expectation. Therefore we show that:

$$\frac{\partial}{\partial\alpha} D_{\mathrm{KL}}(p_\alpha \| p_\theta)\Big|_{\alpha=0} = \lim_{\alpha'\to0^-} \frac{D_{\mathrm{KL}}(p_{\alpha'} \| p_\theta) - D_{\mathrm{KL}}(q_{\mathrm{opt}} \| p_\theta)}{\alpha'} \leq 0. \tag{23}$$

Combining Eq. (22) and (23), we arrive at $\partial D_{\mathrm{KL}}(p_\alpha \| p_\theta)/\partial\alpha|_{\alpha=0} = 0$, which proves the second claim, $H(p_d, q_{\mathrm{opt}}) = H(p_d, p_\theta) - D_{\mathrm{KL}}(q_{\mathrm{opt}} \| p_\theta)$. □

### A.3 PERPLEXITY OF THE OPTIMAL DECODING DISTRIBUTION

The perplexity of the decoding distribution $p_{\theta,\boldsymbol{\mu}}$ can be derived by taking the exponent of the cross-entropy between $p_d$ and $p_{\theta,\boldsymbol{\mu}}$. In practice, we evaluate on the test set $\{\boldsymbol{x}^i\}_{i=1}^N$ which forms an empirical distribution by drawing samples from $p_d$. We also modulate the proposal model by temperature $\tau$ (denoted as $p_\theta^\tau$) to make the result consistent with the inference stage.

We derive the cross-entropy $H(p_d, p_{\theta,\boldsymbol{\mu}})$ at the token-level by plugging in Eq. (2):

$$
\begin{aligned}
H(p_d, p_{\theta,\boldsymbol{\mu}}) &= -\frac{1}{\sum_{i=1}^{N} |\boldsymbol{x}^i|} \sum_{i=1}^{N} \log p_{\theta,\boldsymbol{\mu}}(\boldsymbol{x}^i) \\
&= -\frac{1}{\sum_{i=1}^{N} |\boldsymbol{x}^i|} \sum_{i=1}^{N} \log \left( p_{\theta}^{\tau}(\boldsymbol{x}^i) e^{-E_{\boldsymbol{\mu}}(\boldsymbol{x}^i)} / Z \right) \\
&= -\frac{1}{\sum_{i=1}^{N} |\boldsymbol{x}^i|} \left[ \sum_{i=1}^{N} \left( \log p_{\theta}^{\tau}(\boldsymbol{x}^i) - E_{\boldsymbol{\mu}}(\boldsymbol{x}^i) \right) - \log \sum_{i=1}^{N} \left( p_{\theta}^{\tau}(\boldsymbol{x}^i) e^{-E_{\boldsymbol{\mu}}(\boldsymbol{x}^i)} \right) \right] \\
&= \frac{1}{\sum_{i=1}^{N} |\boldsymbol{x}^i|} \left[ H(p_d, p_{\theta}^{\tau}) + \sum_{i=1}^{N} E_{\boldsymbol{\mu}}(\boldsymbol{x}^i) + \log \sum_{i=1}^{N} \left( p_{\theta}^{\tau}(\boldsymbol{x}^i) e^{-E_{\boldsymbol{\mu}}(\boldsymbol{x}^i)} \right) \right],
\end{aligned}
$$

where $H(p_d, p_{\theta}^{\tau})$ is the cross-entropy of the proposal model with a temperature modulation:

$$
H(p_d, p_{\theta}^{\tau}) = -\frac{1}{\sum_{i=1}^{N} |\boldsymbol{x}^i|} \sum_{i=1}^{N} \sum_{t=1}^{|\boldsymbol{x}^i|} \frac{\exp[\boldsymbol{e}(x_t^i)^{\top} \boldsymbol{h}_t / \tau]}{\sum_{w \in V} \exp[\boldsymbol{e}(w)^{\top} \boldsymbol{h}_t / \tau]}.
$$

where $\boldsymbol{e}(w)$ is the output embedding of $w$ and $\boldsymbol{h}_t$ is the hidden state at time step $t$.

Finally, the perplexity of the decoding distribution $p_{\theta,\boldsymbol{\mu}}$ in DAEMON is $e^{H(p_d, p_{\theta,\boldsymbol{\mu}})}$ in nats.

## A.4 DETAILS OF COEFFICIENTS ESTIMATION WITH WIS

### A.4.1 WIS DERIVATION

We derive the estimation of $\hat{\boldsymbol{F}}$ using Weighted Importance Sampling (WIS) with $N$ i.i.d. trajectories $\{\hat{\boldsymbol{x}}^i\}_{i=1}^{N}$ from the proposal $p_{\theta}$:

$$
\begin{aligned}
\hat{\boldsymbol{F}} &= \mathbb{E}_{\boldsymbol{x} \sim p_{\theta,\boldsymbol{\mu}}}[\boldsymbol{f}(\boldsymbol{x})] \\
&= \sum_{\boldsymbol{x}} p_{\theta,\boldsymbol{\mu}}(\boldsymbol{x}) \boldsymbol{f}(\boldsymbol{x}) \\
&= \frac{\sum_{\boldsymbol{x}} p_{\theta}(\boldsymbol{x}) \exp(-E_{\boldsymbol{\mu}}(\boldsymbol{x})) \boldsymbol{f}(\boldsymbol{x})}{\sum_{\boldsymbol{x}} p_{\theta}(\boldsymbol{x}) \exp(-E_{\boldsymbol{\mu}}(\boldsymbol{x}))} \\
&\approx \frac{\sum_{i=1}^{N} \exp(-E_{\boldsymbol{\mu}}(\hat{\boldsymbol{x}}^i)) \boldsymbol{f}(\hat{\boldsymbol{x}}^i)}{\sum_{i=1}^{N} \exp(-E_{\boldsymbol{\mu}}(\hat{\boldsymbol{x}}^i))}.
\end{aligned}
$$

The last step is obtained by approximating $p_{\theta}(\boldsymbol{x})$ with the empirical distribution defined by $\{\hat{\boldsymbol{x}}^i\}_{i=1}^{N}$, i.e., $\hat{p}_{\theta}^N(\boldsymbol{x}) = \frac{1}{N} \sum_{i=1}^{N} \delta(\boldsymbol{x}, \hat{\boldsymbol{x}}^i)$.

## A.5 DETAILS OF SAMPLING USING SIR

### A.5.1 THE CONDITIONAL FORM OF THE OPTIMAL DECODING DISTRIBUTION

We first start with the auto-regressive factorization of Eq. (2) at step $t$ which marginalizes out the future from step $t$:

$$
p_{\theta,\boldsymbol{\mu}}(x_t | \boldsymbol{x}_{<t}) = p_{\theta}(x_t | \boldsymbol{x}_{<t}) \frac{\mathbb{E}_{\boldsymbol{x}'_{>t} \sim p_{\theta}(\cdot | \boldsymbol{x}_{\leq t})}[\exp(-E_{\boldsymbol{\mu}}(\boldsymbol{x}_{\leq t}, \boldsymbol{x}'_{>t}))]}{\mathbb{E}_{\boldsymbol{x}'_{\geq t} \sim p_{\theta}(\cdot | \boldsymbol{x}_{<t})}[\exp(-E_{\boldsymbol{\mu}}(\boldsymbol{x}_{<t}, \boldsymbol{x}'_{\geq t}))]}.
$$

This form can be derived by breaking down the probability at the last step $p_{\theta,\boldsymbol{\mu}}(x_T | \boldsymbol{x}_{<T})$ and then generalize the derivation to previous steps.

Then the conditional form of $p_{\theta,\boldsymbol{\mu}}$ given prefix $\boldsymbol{x}_{\leq t_0}$ is the product of the per-step probabilities:

$$
\begin{aligned}
p_{\theta,\boldsymbol{\mu}}(\boldsymbol{x}_{>t_0}|\boldsymbol{x}_{\leq t_0}) &= \prod_{t=t_0+1}^{T} p_{\theta,\boldsymbol{\mu}}(x_t|\boldsymbol{x}_{\leq t_0}) \\
&= p_{\theta}(\boldsymbol{x}_{>t_0}|\boldsymbol{x}_{\leq t_0}) \prod_{t=t_0+1}^{T} \frac{\mathbb{E}_{\boldsymbol{x}'_{>t}\sim p_{\theta}(\cdot|\boldsymbol{x}_{\leq t})}[\exp(-E_{\boldsymbol{\mu}}(\boldsymbol{x}_{\leq t},\boldsymbol{x}'_{>t}))]}{\mathbb{E}_{\boldsymbol{x}'_{\geq t}\sim p_{\theta}(\cdot|\boldsymbol{x}_{<t})}[\exp(-E_{\boldsymbol{\mu}}(\boldsymbol{x}_{<t},\boldsymbol{x}'_{\geq t}))]} \\
&= p_{\theta}(\boldsymbol{x}_{>t_0}|\boldsymbol{x}_{\leq t_0}) \frac{\exp(-E_{\boldsymbol{\mu}}(\boldsymbol{x}_{\leq t_0},\boldsymbol{x}_{>t_0}))}{\mathbb{E}_{\boldsymbol{x}'_{>t_0}\sim p_{\theta}(\cdot|\boldsymbol{x}_{\leq t_0})}[\exp(-E_{\boldsymbol{\mu}}(\boldsymbol{x}_{\leq t_0},\boldsymbol{x}'_{>t_0}))]}.
\end{aligned} \quad (24)
$$

The last step is obtained by canceling out the intermediate terms in the product from step $t_0 + 1$ to $T$.

### A.5.2 DERIVING THE SIR APPROXIMATION

The empirical distribution $\hat{p}_{\theta,\boldsymbol{\mu}}^M(\cdot|\boldsymbol{x}_{\leq t_0})$ induced by SIR approximation can be derived from Eq. (24) by substituting the conditional proposal $p_{\theta}(\cdot|\boldsymbol{x}_{\leq t_0})$ with the empirical distribution $\hat{p}_{\theta}^M(\boldsymbol{x}_{>t_0}|\boldsymbol{x}_{\leq t_0}) = \frac{1}{M}\sum_{i=1}^{M}\delta([\boldsymbol{x}_{\leq t_0},\boldsymbol{x}_{>t_0}],\hat{\boldsymbol{x}}^i)$ where $\{\hat{\boldsymbol{x}}_{>t_0}^i\}_{i=1}^M$ are continuation candidates sampled from the conditional proposal $p_{\theta}(\cdot|\boldsymbol{x}_{\leq t_0})$.

$$
\hat{p}_{\theta,\boldsymbol{\mu}}^M(\boldsymbol{x}_{>t_0}|\boldsymbol{x}_{\leq t_0}) = \sum_{i=1}^{M}\delta([\boldsymbol{x}_{\leq t_0},\boldsymbol{x}_{>t_0}],\hat{\boldsymbol{x}}^i) \frac{\exp(-E_{\boldsymbol{\mu}}(\boldsymbol{x}_{\leq t_0},\hat{\boldsymbol{x}}_{>t_0}))}{\sum_{j=1}^{M}\exp(-E_{\boldsymbol{\mu}}(\boldsymbol{x}_{\leq t_0},\hat{\boldsymbol{x}}_{>t_0}))}.
$$

## B RELATED WORK

### B.1 DECODING METHODS

Decoding methods are typically categorized into two main categories: sampling-based methods and search-based methods. Sampling-based methods, which introduce randomness into the selection of next token, are commonly employed in open-ended generation settings to yield diverse texts. Existing sampling-based methods select the next token by sampling from a truncated set of candidates based on heuristics that control the statistics of the next-token probability distribution. For instance, Top-k sampling selects the set of highest $k$ probabilities (Fan et al., 2018), Nucleus sampling focuses on candidates within the $p$-percentile of the highest probabilities (Holtzman et al., 2020), and Typical sampling targets the $\tau$-percentile, which has entropy close to that of the language model (Meister et al., 2022). However, it remains unclear how these controlled statistical quantities correlate with the aspects of generation quality one might concern. In particular, sampling-based methods are often reported to struggle with maintaining topic relevance and long-term coherence. Conversely, vanilla search-based methods, which search for the sequence that maximizes the probability under the language model, tend to produce repetitive and recursively structured texts in open-ended scenarios. Consequently, frequency penalty (Keskar et al., 2019; Dou et al., 2022) is employed to discourage generating tokens that frequently present in the context. Recently, several works have been proposed to maximize contrastive objectives. Contrastive Decoding (Li et al., 2022) seeks the token that maximizes the probability difference between an expert LM and an amateur LM. Contrastive Search (Su et al., 2022) searches for the token that maximizes the probability given by the LM while minimizing its representation similarity to the previous tokens. Nevertheless, the output of these methods still frequently exhibits redundancy in the long term due to the intrinsic bias in the language model, despite the aim of their objectives to reduce semantic repetition. Our method, categorized as a sampling-based method, aims to induce the optimal decoding distribution that explicitly aligns with the underlying distribution of human text, as assessed by evaluation metrics related to a set of chosen aspects.

### B.2 LANGUAGE MODEL TRAINING METHODS FOR QUALITY IMPROVEMENT

Prior works also attempted to improve the generation quality by devising new training objectives to further train the language model. One approach involves the design of auxiliary objectives aimed

at discouraging the model from learning implausible samples or features by the standard Maximum Likelihood Estimation (MLE) objective. Welleck et al. (2020) proposed unlikelihood training objective that directly penalizes token-level and phrase-level repetition. Xu et al. (2022) proposed the DITTO objective that penalizes sentence-level repetition loop. Su et al. (2022) proposed a contrastive objective that separates the representations of distinct tokens in the context to facilitate decoding with repetition penalty. Despite the direct mitigation of undesired behavior of the model, these objectives are usually applied to the local probability of next token prediction by the language model, which is conditioned on the ground-truth contexts. This leads to a discrepancy between training and inference, i.e., the exposure bias, raising concerns regarding the preservation of the learned properties at the inference stage. Another research direction involves the application of Reinforcement Learning (RL) to optimize the generation model towards sequence-level metrics (Ranzato et al., 2016; Shen et al., 2016; Yu et al., 2017; Guo et al., 2018; Shi et al., 2018). Although targeting at optimizing the model at a sequence level, most RL approaches often underperform MLE especially with a pre-trained base model. Pang & He (2021) demonstrated the effectiveness of off-policy RL that learns from human demonstration with quality-centric reward over the approach of direct fine-tuning. More recent research explored learning the reward function from human judgment (Jaques et al., 2019; Ziegler et al., 2019; Ouyang et al., 2022) and has shown to promote the pre-trained language model to generate texts whose quality is more aligned with human preferences. Our method circumvents exposure bias and the challenges associated with RL optimization by deriving the analytic formulation of the proposed optimization problem, which directly aligns the expected metric scores of generated texts with those of human texts.

### B.3    ENERGY-BASED MODEL FOR TEXT GENERATION

The Energy-Based Model (EBM) (Hinton, 2002; LeCun et al., 2006) is a generative model that learns the underlying distribution by relocating energy based on sampled data points. In text modeling, the EBM is attractive due to its global sequence scoring, as opposed to the locally normalized score factorization in Auto-Regressive (AR) models (Rosenfeld et al., 2001; ROSENFELD, 1996). Theoretical work has demonstrated that the EBM family offers greater expressiveness than the AR model family by encompassing a broader range of computable text distributions (Lin et al., 2021a). However, two critical aspects of the EBM, namely, learning and sampling, pose challenges for its practical application in text generation. While AR models can be trained by directly maximizing the likelihood of reference data, learning a parametric EBM typically involves Noise-Contrastive Estimation (NCE) (Gutmann & Hyvärinen, 2012; Ma & Collins, 2018). With the recent advances in pre-training, previous studies proposed to construct the EBM based on a pre-trained language model with an exponential energy term that is parametrized by a neural network (Deng et al., 2020) or a log-linear model (Parshakova et al., 2019a). Specifically, the latter form emerges as the solution based on the generalized maximum entropy principle (or minimum discrimination information principle), incorporating linear constraints on model expectations (see Eq. (1)), which is also explored in Posterior Regularization technique (Ganchev et al., 2010). This formulation has also found applications in areas such as controlled text generation with distributional control (Khalifa et al., 2021) and calibrating the entropy rate of language models (Braverman et al., 2020), among others. On the other hand, achieving scalable and tractable sampling from the EBM has long been a challenge, primarily because of its globally normalized form. Traditional MCMC approaches such as Gibbs Sampling (Geman & Geman, 1984) or Metropolis Hastings (Metropolis et al., 1953) often face scalability issues concerning data dimensionality and model complexity. Recent endeavors (Parshakova et al., 2019b; Khalifa et al., 2021) proposed to learn another AR policy guided by the EBM. Nevertheless, this approach compromises the modeling capacity of the EBM, as it involves learning an AR policy that minimizes the forward Kullback-Leibler (KL) divergence towards the solution of the optimization problem. Finally, energy-based reranking methods (Bhattacharyya et al., 2021; Fernandes et al., 2022; Freitag et al., 2022) are explored in machine translation where the generations are reranked according to pre-defined reference-based metrics. However, these methods cannot guarantee to improve the distribution of the base generation model to resemble the underlying text distribution from humans. They only focused on maximizing certain quality metrics while neglecting various aspects of human texts (e.g., quality, diversity, repetition and etc.) which are necessary for achieving **human-like generation**. Our approach generalizes the constraint functions to encompass evaluation metrics across various aspects in the text decoding formulation with the goal of alignment with human texts, and facilitates tractable sampling from the EBM by leveraging a strong AR proposal

through Sampling-Importance-Resampling (SIR), which possesses a well-defined convergence rate.

## C  DISCUSSION ON THE FORWARD AND REVERSE KL DIVERGENCE

We provide an in-depth discussion on the forward and reverse KL divergence. We reuse the notation in Section 2.1, and denote the distribution induced by a language model as $p_\theta$ and the decoding distribution as $q$. Although both the forward and the reverse KL divergence can be used to measure the distance between $q$ and $p_\theta$, they are not symetric (Bishop & Nasrabadi, 2006), i.e., $D_{\mathrm{KL}}(p_\theta\|q) \neq D_{\mathrm{KL}}(q\|p_\theta)$, and have distinct differences in shaping the decoding distribution.

Minimizing the reverse KL divergence, $D_{\mathrm{KL}}(q\|p_\theta)$ is known to encourage **zero-forcing** behavior (Malinin & Gales, 2019), as it forces $q(x) = 0$ when $p_\theta(x) = 0$. This restricts the decoding distribution $q$ to be **mode-seeking** and only explores the modes of the target distribution $p_\theta$ which contains samples with high likelihood under the ground-truth model's distribution $p_\theta$, and thus ignores the outliers.

Whereas, minimizing the forward KL divergence, $D_{\mathrm{KL}}(p_\theta\|q)$ encourages **zero-avoiding** behavior, i.e., it avoids $q(x) = 0$ when $p_\theta(x) > 0$. This leads $q$ to be **mean-seeking** which spreads its support to cover all the non-zero density regions of the model distribution $p_\theta$. As the distribution of language models are known to have unreliable long tails (Holtzman et al., 2020), the decoding distribution that minimizes the forward KL can further overestimate those tails to produce low-quality samples (Chan et al., 2022).

Overall, comparing to the forward KL, minimizing the reverse KL leads to a more focused distribution and ignores the long tail in the given language model distribution, which is more suitable for our decoding scenario that demands generation quality. We also add an intuitive visualization to illustrate the different behaviors of reverse and forward KL in Figure 4.

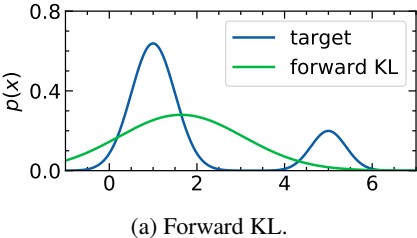
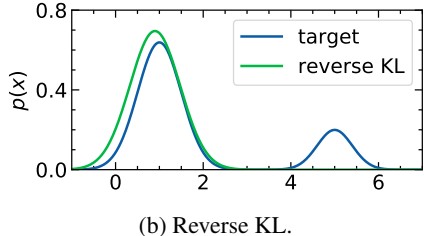

(a) Forward KL.                    (b) Reverse KL.

Figure 4: The mean-seeking behavior of forward KL and the mode-seeking behavior of reverse KL.

## D  HUMAN EVALUATION SETUP

### D.1  ANNOTATION PROCESS

We conduct pair-wise human evaluation with Amazon Mechanical Turk workers, providing them with comprehensive instructions and detailed explanations of annotation features. To ensure that participants can fully comprehend and make accurate judgments, we conduct a pilot qualification test using 30 pairs. For the formal annotation process, we randomly select 100 prefixes, each with a length of 32, from the test set of Wikipedia. We employ GPT-2 XL (1.5B) with four baseline decoding methods (i.e., contrastive decoding, contrastive search, nucleus sampling, and typical decoding) and our proposed DAEMON algorithm to generate text continuations, each with a maximum length of 256.

Each Human Intelligence Task (HIT) consists of the prefix, the two continuations, and evaluation questions regarding three evaluation criteria. The HITs are randomly shuffled and evaluated by three annotators, adding up to 1,200 annotation samples (100 prefixes × 4 baselines × 3 criteria). Annotators are required to assess and choose the better continuation or opt for a draw according to three evaluation criteria: coherence, fluency, and informativeness. Annotators are paid at a rate of $0.5 per HIT.

Participation in the annotation task is exclusively open to qualified workers that meet the following qualifications:

- HIT Approval Rate must be greater than 95%.
- Number of HITs Approved must exceed 500.

For quality control, we conduct manual cross-validation to check the annotation results based on the following rejection principles. The unqualified HITs are rejected and requested re-annotation.

- Review time is less than 500s.
- Obvious Misjudgment. Key points include: (i) Assigned higher fluency to the text with more grammar mistakes or repetition. (ii) Assigned higher coherence to the text with more jumbled topics. (iii) Assigned higher informativeness to the text with more redundant expressions.

### D.2 ANNOTATION GUIDELINES

**Instruction:** We will give a short prefix and two continuations generated by two systems. You are required to compare the *fluency*, *coherence*, and *informativeness* of two continuations based on the detailed guidelines. The three annotation features are explained as follows:

**Coherence:** Coherence assesses the semantic consistency of the generated text with the prefix. Key aspects encompass:

- **Discourse Coherence:** The generated text maintains semantic connections and appropriate transitions.
- **Logical Coherence:** The generated text maintains logically consistent and avoids self-contradiction (e.g., chronological, cause-and-effect, narrative).
- **Global Coherence:** The generated text aligns with the provided prefix and remains the same topic, event, or entities.

**Fluency:** Fluency refers to the extent to which the generated text is smooth, highly readable, and easily comprehensible. In contrast to coherence, fluency primarily underscores intra-sentence readability. Several pivotal facets encompass:

- **Grammar:** Grammatical correct and adheres to proper English grammar.
- **Structure:** Clarity in sentence structure.
- **Semantic:** Absence of repetitions or omissions.
- **Vocabulary:** Avoidance of inappropriate phrase combinations or symbols.

**Informativeness:** Informativeness refers to the richness and diversity of generated text. Key aspects include:

- **Comprehensive Information:** The text should offer valuable information with engaging details, avoiding redundancy and triviality.
- **Lexical Diversity:** Use varied vocabulary and sentence structures.
- **In-depth Details:** Provide pertinent, in-depth details.
- **Novelty:** Introduce new elements while maintaining *coherence*.

## E  RUNTIME ANALYSIS OF ALGORITHM 1

We first analyze the computational cost of Algorithm 1, which mainly consists of two parts.

The first part takes one run of unconditional sampling from the base language model (line 2 of Algorithm 1). We typically set the number of samples to be the same as the development set, so that this cost is equal to one time sampling on the development set.

The second part is estimating the parameters $\{\mu_i\}_{i=1}^K$ by iterative gradient descent (line 3-6 of Algorithm 1). As the number of parameters is small (equals to the number of constraints which is usually in the scale of dozens in practice), and the metric scores $\{f_i(x)\}_{i=1}^K$ can be pre-computed, the total computational overhead is very small. In our experiment, it takes less than one minute to perform thousands of gradient descent steps, which can be ignored comparing to the cost in sampling from the language model.

Comparing to typical grid search of hyper-parameters which requires $H \times R$ runs of sampling in the development set, where $H$ stands for the number of hyper-parameters and $R$ is the number of trials for each hyper-parameter. Even the labor intensive manual tuning methods require at least one run of sampling on the development set, which do not have any advantage over our Algorithm 1 in terms of computational cost.

## F   COMPLEXITY AND RUNTIME ANALYSIS OF DECODING METHODS

We provide analysis about the computational complexity and the runtime in practice of different decoding methods and our method (Algorithm 2). In the following, we consider the case of generating a completion with full length $T$ given an input prompt with length $t_0$ (much smaller than $T$). To evaluate the real runtime performance, we follow the setting in Section 3.6 which uses GPT2-XL to generate completions with maximum length of 256 on the test set of Wikitext. The experiment was done on a Tesla V100.

**Greedy Decoding.** At each decoding step, the Transformer model attends to the previous tokens with the complexity of $O(T)$ and then performs positional transformation, e.g., linear mapping, layer normalization and etc., which is not dependent on $T$. Hence, the full complexity of generating the entire completion is $O(T^2)$. As greedy decoding only picks the token with the maximum probability, it does not incur additional complexity at each decoding step. In our experiments, the runtime for decoding a completion using greedy decoding is 9.33 seconds on average.

**Top-$k$ Sampling.** Top-k sampling has nearly the same complexity as greedy decoding. The only difference is that at each step it samples from the subset that has top-$k$ probabilities rather than taking the maximum. In our experiments, the runtime for decoding a completion using top-$k$ decoding is 9.34 seconds on average, which is almost the same as greedy decoding.

**Nucleus Sampling.** At each decoding step, Nucleus sampling sorts the probability distribution, which incurs an additional complexity of $O(V \log V)$ where $V$ is the vocabulary size comparing to greedy decoding or top-k sampling. In our experiments, the runtime for decoding a completion using nucleus decoding is 10.54 seconds on average, which is 1.13 times of the latency of the greedy decoding.

**Contrastive Decoding.** Contrastive decoding uses two language models with different sizes for decoding, where a single language model has the complexity of $O(T^2)$. During decoding, they use beam search which increases the total forward complexity to $O(BT^2)$ for a beam size of $B$. Additionally, at each decoding step, beam search has to sort the probabilities in the beams which has a complexity of $O(BV \log(BV))$. In our experiments, we followed the beam size of 5 suggested inLi et al. (2022), and the runtime for decoding a completion using contrastive decoding is 12.10 seconds on average, which is 1.29 times of the latency of the greedy decoding.

**Contrastive Search.** At each decoding step, contrastive search has to perform two forward passes of the language model. After the first forward pass, the algorithm selects the top-$k$ candidates and performs the second forward pass to get the hidden representations of these candidates. This increases the per-step complexity to $O(T)+O(kT)$, which results in the total complexity of $O(T^2)+O(kT^2)$. Note that the two forward passes cannot be parallelized and the serialized execution slows down the run time. In our experiments, we set $k$ to $5$ according to Su et al. (2022), and the runtime for decoding a completion using contrastive search is 11.82 seconds on average, which is 1.27 times of the latency of the greedy decoding.

**DAEMON (Ours).** Our decoding method DAEMON has two stages, i.e., candidate sampling and re-sampling to approximate the optimal sampling distribution. In the first sampling stage, we sample $M$ candidate sequences in parallel, whose total complexity is $O(MT^2)$. In the second re-sampling stage, we compute energy for each candidate sequence and re-sample from the distribution defined

by the energy. The second stage has a complexity of $O(CM)$ where $C$ denotes the complexity of calculating the energy, which is much smaller than the cost of sampling. Note that unlike contrastive decoding and contrastive search, in the first sampling stage of our method, sampling $M$ sequences can be fully parallelized, which is highly optimized by modern GPU hardware. Hence the actual inference latency grows much slower when increasing $M$. In our experiments, we set $M$ to 25, and the runtime for decoding a completion using DAEMON is 12.58 seconds on average, which is 1.35 times of the latency of the greedy decoding.

## G    CONVERGENCE AND SENSITIVITY ANALYSIS OF ALGORITHM 1

We analyze the convergence of $\boldsymbol{\mu}$ in Algorithm 1 using different initializations of $\boldsymbol{\mu}$. Concretely, we consider the following three initializations:

- `zero`: Initializing all dimensions of $\boldsymbol{\mu}$ to 0.
- `randn`: Initializing dimensions of $\boldsymbol{\mu}$ with random numbers sampled from a standard normal distribution $\mathcal{N}(0, 1)$.
- `rand`: Initializing dimensions of $\boldsymbol{\mu}$ with random numbers sampled from a uniform distribution $U[0, 1]$.

We run Algorithm 1 and consider the following nine metrics in the main experiment: SEQ-REP-2, SEQ-REP-3, SEQ-REP-4, TOK-REP-8, TOK-REP-16, TOK-REP-32, COH, DIV, $e^{\text{ENT}}$ on Wikitext. Each $\mu_i$ ($i \in \{1, \ldots, 9\}$) corresponds to the above metric respectively.

We plot the optimization trajectory of each $\mu_i$ under Algorithm 1 with different initializations in Figure 5. We took 5 runs with different random seeds for each initialization.

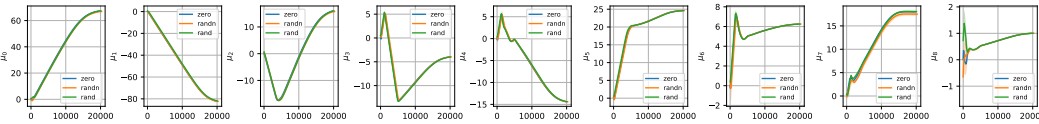

Figure 5: Optimization curve of each $\mu_i$ with three different initializations: `zero`, `randn`, `rand`.

From the result, we observe that the optimization of $\boldsymbol{\mu}$ is quite stable and all the trajectories converge to almost the same optimal solutions under different initializations.

## H    ROBUSTNESS ANALYSIS

We study the robustness of the alignment results on different metrics when one chooses to optimize a single metric. We conduct our experiments using GPT-2 XL on Wikitext and perform a grid search with different combinations of $M \in \{10, 25, 50, 100\}$ and $\tau \in \{0.96, 0.97, 0.98, 1.0\}$ to optimize towards aligning a single metric in one of the following metrics: SEQ-REP-4, TOK-REP-32, COH, DIV, $e^{\text{ENT}}$. From the result in Table 5, we observe that the performance on different metrics has a very small variance even when we optimize for a single metric, which indicates the robustness of DAEMON.

## I    ADDITIONAL RESULTS ON TEXT SUMMARIZATION

To demonstrate the universability of our method, we conducted an experiment on the standard summarization benchmark CNN/DailyMail using an off-the-shelf summarization model `pegasus-cnn_dailymail`[3]. As shown in Table 6, compared with the baseline sampling methods, our method achieves better performance in aligning with most aspects including repetition (SR-3, TR-8), coherence (COH), diversity (DIV) and information ($e^{\text{ENT}}$) (better when the score is closer to that of the reference), and ROUGE scores (better when the score is higher).

---

[3] https://huggingface.co/google/pegasus-cnn_dailymail

| Model | SR-4 | TR-32 | COH | DIV | $e^{\text{ENT}}$ |
|---|---|---|---|---|---|
| Reference | 0.48 | 21.3 | 62.3 | 92.5 | 23.2 |
| opt. SR-4 | 0.42 | 22.5 | 62.5 | 92.2 | 22.8 |
| opt. TR-32 | 0.38 | 21.2 | 62.4 | 94.1 | 24.3 |
| opt. COH | 0.38 | 21.2 | 62.4 | 94.1 | 24.3 |
| opt. DIV | 0.55 | 22.9 | 63.3 | 92.7 | 22.2 |
| opt. $e^{\text{ENT}}$ | 0.40 | 21.5 | 61.6 | 93.9 | 23.1 |

Table 5: Quality of generation measured under all metrics when one only optimizes for a single metric at decoding time. For instance, "opt. SR-4" means only optimizing SR-4.

| Model | SR-3 | TR-8 | COH | DIV | $e^{\text{ENT}}$ | R-1 | R-2 | R-L |
|---|---|---|---|---|---|---|---|---|
| Reference | 0.10 | 2.93 | 80.1 | 99.1 | 31.2 | - | - | - |
| Top-k | 0.15 | 3.10 | 79.6 | 99.2 | 35.0 | 38.7 | 15.0 | 35.6 |
| Nucleus | 0.15 | 3.00 | 72.3 | 99.0 | 104.3 | 34.4 | 12.2 | 31.6 |
| Typical | 0.11 | 2.71 | 76.3 | 99.2 | 81.8 | 34.3 | 12.1 | 31.5 |
| CS | 0.15 | 3.14 | 80.9 | 98.8 | 28.4 | 40.9 | 16.9 | 37.8 |
| Daemon | **0.10** | **2.98** | **80.1** | **99.2** | **29.7** | **41.9** | **18.4** | **38.9** |

Table 6: Results on the CNN/DailyMail.

## J    EXPERIMENT DETAILS

### J.1    STATISTICS OF DATASETS

| Domain | # Dev set | # Test set | Prefix length | Full length |
|---|---|---|---|---|
| Wikipedia | 512 | 802 | 24.8 | 183.1 |
| News | 512 | 1488 | 25.6 | 309.5 |

Table 7: Statistics of the data used in the experiments.

The data statistics of each domain are summarized in Table 7. For each example, the first 32 words are used as the prefix. Each word is tokenized into subwords before fed into the embedding layer of the language models. Both GPT-2 and OPT use Byte-Pair Encoding subword segmentation. The average length of subwords in the prefix is around 32. The maximum length of subwords in the complete generation is restricted to 256. During evaluation, the subwords length of human references is also truncated to 256 for a reasonable comparison.

### J.2    IMPLEMENTATION DETAILS

We first describe the implementation details of the baselines. We first consider three canonical sampling-based methods: **Top-k** sampling (Fan et al., 2018) samples from the set of candidates with top-$k$ probabilities. **Nucleus** sampling (Holtzman et al., 2020) samples from the set of $p$-percentile that has minimum number of candidates. **Typical** decoding (Meister et al., 2022) samples from the set of $\tau$-percentile whose entropy is close to the entropy of the LM. For search-based methods, besides vanilla **Greedy** decoding, we also consider two recent methods that maximize contrastive objectives: Contrastive Decoding (**CD**) (Li et al., 2022) searches for the token that maximizes the probability difference between an expert LM and an amateur LM scaled with temperature $\tau$ within a candidate set controlled by $\alpha$. Contrastive Search (**CS**) (Su et al., 2022) searches for the token that maximizes probability given by the LM while minimizing its representation similarity to the previous tokens with an intensity of $\alpha$ within the top-$k$ candidate set. For all baselines, we use the recommended hyper-parameter settings in the original papers. Specifically, we use $k = 50$ for Top-k sampling, $p = 0.95$ for Nucleus sampling, and $\tau = 0.95$ for Typical decoding. For

| Model | SR-2 | SR-3 | SR-4 | TR-8 | TR-16 | TR-32 | COH | DIV | $e^{\text{ENT}}$ | MAU |
|---|---|---|---|---|---|---|---|---|---|---|
| Reference | 5.82 | 1.40 | 0.48 | 5.77 | 12.8 | 21.3 | 62.3 | 92.5 | 23.2 | - |
| Greedy | 66.2 | 62.9 | 60.9 | 13.5 | 39.4 | 65.5 | 60.3 | 8.03 | 2.29 | 59.7 |
| Top-$k$ | 8.48 | 3.28 | 2.11 | 6.74 | 14.5 | 23.4 | 60.9 | 87.8 | 10.1 | 77.8 |
| Nucleus | **5.51** | 1.84 | 1.19 | **5.98** | **12.4** | 20.0 | 57.3 | 92.4 | 17.3 | 78.3 |
| Typical | 3.98 | 1.24 | 0.81 | 5.07 | 10.7 | 17.4 | 54.9 | 94.5 | 30.1 | 78.7 |
| CD | 10.5 | 3.00 | 1.31 | 8.00 | 17.3 | 28.2 | 68.7 | 86.0 | 7.55 | 77.8 |
| CS | 6.71 | 2.51 | 1.78 | 6.40 | 14.1 | 23.0 | 56.9 | 90.6 | 5.25 | 83.3 |
| DAEMON | 6.28 | **1.31** | **0.42** | 6.59 | 13.6 | **22.5** | **62.5** | **92.2** | **22.8** | **88.1** |
| Greedy | 61.4 | 57.3 | 54.8 | 12.6 | 35.5 | 60.4 | 62.0 | 12.6 | 2.78 | 64.8 |
| Top-$k$ | 9.14 | 3.77 | 2.44 | 7.33 | 15.1 | 24.1 | 61.3 | 86.6 | 13.9 | 77.5 |
| Nucleus | 7.69 | 3.36 | 2.33 | 6.48 | 13.5 | 21.9 | 59.1 | 88.6 | 18.9 | 80.1 |
| Typical | 5.16 | 1.70 | 1.06 | **5.70** | **12.2** | 19.6 | 57.0 | 92.9 | 31.9 | 77.7 |
| CD | 11.8 | 4.90 | 2.92 | 6.81 | 15.3 | 26.5 | 68.6 | 82.3 | 11.7 | 78.6 |
| CS | **5.89** | 1.94 | 1.14 | 6.32 | 13.7 | 21.7 | 57.7 | 91.8 | 8.72 | 83.3 |
| DAEMON | **5.89** | **1.33** | **0.38** | 6.19 | 13.3 | **21.6** | **62.3** | **92.6** | **22.7** | **90.7** |

Table 8: Full main results of automatic evaluation on the Wikipedia domain using GPT-2 XL and OPT-6.7B. For all metrics, the best scores (in boldface) are the *closest* to the human scores except for MAU, which is better when *higher*.

contrastive decoding, we directly use the generated texts provided by the official implementation[4] with hyperparameter setting: $\alpha = 0.1$, $\tau = 0.5$ for GPT-2 and $\tau = 1.0$ for OPT. For contrastive search, we follow the official implementation on the datasets (Su & Xu, 2022) that uses $\alpha = 0.6$, $k = 5$.

For our method DAEMON in the main results, we use the following metrics in the constraints: SR-2, SR-3, SR-4, TR-8, TR-16, TR-32, COH, DIV, $e^{\text{ENT}}$ described in §3.2. For coefficients estimation, we estimate the target expectation on the dev set and fit the coefficients using Adam with learning rate 5e-3 until a minimum error 1e-3 is reached. The inference of DAEMON with the base model of either GPT-2 XL or OPT-6.7B can be done on a Tesla V100 with a batch size of 1.

## J.3 FULL MAIN RESULTS

We present the full results of all metrics on the Wikipedia and News domain in Table 8 and Table 9 respectively.

## J.4 QUALITATIVE CASES

We present qualitative cases generated by four baselines and DAEMON in Table 10, 11, 12, 13, 14, 15.

---

[4]`https://github.com/XiangLi1999/ContrastiveDecoding/`.

| Model | SR-2 | SR-3 | SR-4 | TR-8 | TR-16 | TR-32 | COH | DIV | $e^{\mathrm{ENT}}$ | MAU |
|---|---|---|---|---|---|---|---|---|---|---|
| Reference | 4.76 | 0.93 | 0.29 | 4.72 | 10.8 | 18.7 | 66.6 | 94.1 | 13.8 | - |
| Greedy | 58.7 | 55.1 | 53.2 | 8.06 | 28.0 | 58.2 | 63.8 | 13.2 | 2.19 | 65.2 |
| Top-$k$ | 6.16 | 1.80 | 0.95 | 5.26 | 11.8 | 20.3 | 64.7 | 91.7 | 8.17 | 96.3 |
| Nucleus | 4.91 | 1.39 | 0.80 | 4.93 | 10.9 | 18.7 | 60.8 | 93.5 | 11.0 | 95.3 |
| Typical | 3.62 | 0.83 | 0.42 | 4.50 | 9.98 | 16.9 | 57.2 | 95.3 | 18.2 | 95.0 |
| CD | 7.45 | 1.78 | 0.63 | 5.97 | 13.5 | 23.2 | 71.2 | 90.5 | 6.55 | 95.1 |
| CS | 4.45 | 1.23 | 0.77 | 4.68 | 11.0 | 19.2 | 63.6 | 94.1 | 4.18 | 95.7 |
| DAEMON | **4.64** | **0.71** | **0.18** | **4.71** | **10.8** | **18.7** | **66.3** | **94.5** | **13.7** | **97.4** |
| Greedy | 51.3 | 47.2 | 45.2 | 7.75 | 24.5 | 51.0 | 63.6 | 21.9 | 2.72 | 70.7 |
| Top-$k$ | 6.73 | 2.42 | 1.53 | 4.93 | 11.4 | 19.9 | **65.7** | 90.5 | 10.7 | 95.7 |
| Nucleus | 6.03 | 2.13 | 1.37 | 4.95 | 11.1 | 19.2 | 63.3 | 91.5 | 12.2 | 95.3 |
| Typical | 4.59 | 1.47 | 0.95 | 4.52 | 10.1 | 17.7 | 59.4 | 93.7 | 19.4 | 95.2 |
| CD | 8.34 | 3.23 | 1.94 | **4.91** | 11.5 | 21.0 | 71.7 | 87.5 | 9.20 | 95.2 |
| CS | **4.74** | 1.74 | 1.18 | 4.52 | 10.6 | 18.3 | 62.9 | 93.2 | 6.69 | 94.0 |
| DAEMON | 4.30 | **0.67** | **0.25** | 4.52 | **10.7** | **18.8** | 64.8 | **94.9** | **13.6** | **97.2** |

Table 9: Full main results of automatic evaluation on the News domain using GPT-2 XL and OPT-6.7B. For all metrics, the best scores (in boldface) are the *closest* to the human scores except for MAU, which is better when *higher*.

| | |
|---|---|
| *Prefix* | Morris Dickstein of The New York Times was more critical of the book, calling it " a misconceived and badly executed novel, a sideshow distraction from |
| **CD** | a much bigger story " and saying the book's author had failed " in her most basic duty to the public." The Times review, by David Streitfeld, is here and the book review by Morris Dickstein, by the Times, here. The review by Morris is more critical, calling it "a misconceived" and " badly executed novel," a sideshow distraction from a " much bigger" story and calling the author a failure "in" her "basic" duties. The review by David is more positive. He says that it "has its charms, and the author is clearly a talented story teller," and he praises the author for "her skill in weaving an engaging tale." The book is available here. The book by Morris is more critical. He calls the novel a sideshow distraction and a "misconceived and" a badly "imperfectly" executed novel, a "fails in its most basic" duties |
| **CS** | the main event." In an interview with the Los Angeles Times, Morris Dickstein said, "There's no reason to write a book about this that's going to get people's attention except to make it into a movie, and I'm not a movie person at all. The idea of making a movie out of this is a bad idea in my opinion." Morris Dickstein's comments were met with derision by a number of authors, including David Mamet, who wrote in his review of the book, "It's hard to imagine a writer more in the vanguard of the literary avant-garde than Morris S. Dickstein, whose work is the stuff of literary legend: The Man Who Knew Too Much, A Passage to India, All Quiet on the Western Front, To Kill a Mockingbird to name a few of his masterpieces." Mamet went on to say, "Dick |
| **Typical** | its own lame screenplay" while reviewing edition number 632, 1952. Here's an excerpt from an actual phone interview that took place between Normen's editor, Margaret Woods Hamlin, and a reporter from Life magazine. You can try remaining objective when it comes to incompetent film scripts, then. Madame Normen in 1952 with one of the terrible terrible script-writing venues where she allowed herself to stand and present like an academy award nominee (the anything can be anything, anybody can fly, anybody can speak ad lib, "These words and ideas can put a tray of mutton bones into a mouse's muffin; to me they mean nothing.") Princess Ping-Pong: What do you mean you hate that book? Should I deny it? Normen: That book is re-naming ballet for beginners. Do you know, one of the earliest posters coming from front pages in the finest role they can find |
| **Nucleus** | 22900 and One Hundred Thousand Grapes. In 2010, he published a book called The Disappearance of the Jews in Europe: A Statistical History, and it just came out today. In this episode, which makes for a very odd sort of audio, A.O. Scott takes over the show and discusses the Holocaust in nearly eight minutes, describing the era, its mood, the statistics, its victims, its symbols, the orchestrating forces, and even some of its realities. All of it is chilling, really, coming from a man who was born one month after the holocaust ended. Best listen: Enjoy the full minute and twenty-five seconds if you can. This week, we've got an exclusive preview of a whole new book from George Steiner, on the Holocaust: www.housedistory.org/horusmacherbooks. You can also hear A.O. Scott and I |
| **DAEMON** | the questions of it's own dubious existence, and a carbon copy of a similarly flawed book that deals with the woman the author has always hated – her mother – in a more kind and humble way." A correct assessment of the novel as a book beats an unduly critical one quoted a correctly dismissive one. The book is good – it's an intelligent and emotional account of the social influence of the publishing industry on children's books, part of Dickstein's ongoing concern with capitalist and legal reform. Dickstein doesn't spoon-feed the reader plot points and he allows the reader to create her own meaning in constructing the novel's narrative. The author doesn't wander into the busy life of a teenager, which we remember often enough. The opinions of our readers are important to us – just click on one of these and pass along to friends. This opinion article was written by an independent writer. The opinions and views expressed herein |

Table 10: Randomly selected generations from the test set of the Wikipedia domain.

| | |
|---|---|
| *Prefix* | Chad is a landlocked country in Africa whose northern region lies within the eastern reaches of the Sahara Desert and whose southern region lies within the eastern portion of the |
| **CD** | Arabian Desert. Chad's capital city is Ndjamena and it borders the Republic of Mali on its west, Sudan to its north, the Central Sudanese Republic to the north-east, the Central Sahel on its eastern side and the Republic of Nigeria in the east. Chad is bordered to its south-west and east by Cameroon, Sudan and Libya, while its west by the Republic of Chad and to its north by Algeria. Chad shares a land boundary with Sudan to its west and with Libya and Mali on the north and west respectively. Chad's population of approximately 11 million people are mainly Christian, while a small Muslim population exists. Chad's economy is largely agro-based with agriculture and cattle raising the major economic activities, while mining is also a significant economic sector. Chad's main export products include cocoa and oil palm, while its major import is crude petroleum and refined oil products. Chad's main natural resource are its mineral deposits, including |
| **CS** | Sahel.
Chad has a population of approximately 18 million, of which more than half are ethnic Tuareg and Berber populations. In recent years, there has been a rise in Islamist extremism in the country, which has contributed to the country's security situation. The government has taken steps to counter this threat, including the deployment of military forces to the border with Mauritania and the use of air strikes against al-Qaeda-linked militants in the north of the country.
Terrorism Threat Worldwide - 2014 World Terrorism Report (WT) The 2014 Worldwide Threat Assessment (WT) from the Combating Terrorism Center at West Point, New York, assesses that the threat from transnational terrorist groups remains high. The report, based on data from January through December 2014, identifies four groups that pose the greatest threat to the United States: ISIL (Islamic State of Iraq and the Levant), Boko Haram, Jemaah Islamiyah (JI |
| **Typical** | Great Rift Valley. Chad founded the Empire of the Camels sometime after the reign of Muhammad's grandson Hassan II.
Contents show]
Geography Edit
Chad is one of the three remaining provinces in Africa that lies under the jurisdiction of the Holy Roman Empire. Vast swathes of territory are under the dominion of the Kingdom of Niger, although as a unified independent country, their military is limited to a single regiment of regular infantry and reserves of trained cavalry. Its evaluation rate is so low that the Realm Style of the western continent is the standard of the Kingdom. The southern border lies on Rivers of Gas and Darell. During the 20th Century, Chad was experianced the devastating civil war known as the Azawad War with the Kingdom of the Camels. Its long-range artillery is inferior to the Kingdom's regiment of heavy artillery—dark-reddish guns. An earldom-based system of lands are spread |
| **Nucleus** | Great Rift Valley. Chad is bordered by the Ethiopian border. Chad is located on the north-west edge of the Sahara Desert and is located within the region of the Great Rift Valley.
Language [ edit ]
The official languages in Chad are Standard Arabic, Coptic, and Rambani. Other major tongues include the Acholi languages of the Amhara region, the Jolof languages, and the Mboik languages of the Niger Delta.
Names [ edit ]
Chad has six official family names: a primary name, the paternal name and the maternal name, the two father's first names, and the mother's maiden name. For most people the whole family name is usually the same. If a man leaves his surname and his father's name, and is later married, his new surname is his husband's family name in addition to his husband's initial. An assumption of the mother's name by a man |
| **Daemon** | Sahel Region of Africa, with a temperate rainless climate and abundant rainfall. Chad has quite a few diverse altitudes, including the mountain ranges of the Karo, Sawman, Kalun, and Chambek Mountains. The tallest mountain in Chad is the 2900 meter Karo Mountain, which is characterized by its relative dryness and dry season during the dry season.
For more than 600 years, Chad has been a part of Mali, but Mali was only recognized as a country in 1960 for political reasons. Sebbi and nearly all of Lake Chad are also officially part of Mali. Chad's population is of around 8 million, with approximately 85% of the population living in the capital, Ndjamena. The population is also heavily dependent upon industry and agriculture and employs over 50 million people nationally. Per capita income has remained fairly steady for decades, and labor is a hot-topic of debate with labor unions, advocacy organizations, |

Table 11: Randomly selected generations from the test set of the Wikipedia domain.

| *Prefix* | Lesnar's rapid rise to the top of WWE in 2002 led to a feud with The Undertaker, which involved a match at Unforgiven. |
|---|---|
| **CD** | Lesnar defeated The Dead Man in the match and became the WWE Champion, but The Deadman would return the following night to defeat him for his second WWE Championship reign, and the feud continued on for years, culminating in a match at Survivor Series 2002, which The Undertaker won in an incredible fashion, with a Tombstone Piledriver. The match was so good, it was later included as a DVD extra in WWE: Back in Black. Lesnar's next match after Unforgiven, at Summer Slam 2003 against Kurt Angle and Shawn Stasiak, would also end in an Angle pin. The next match he had after Summer Slam, at No Way Out, saw him defeat Edge, who he had previously beaten for the World Championship at Wrestlemania X8, to become the first ever Triple Crown Champion. He would then defeat Triple H at No Mercy to retain his titles for the first and only time in the company, but lose them at Backlash to Batista |
| **CS** | WWE Hall of Famer Jim Rosswas ringside for the match, and told Sports Illustrated in a story published last week that he had to get his camera out of his pants to take a picture of the two superstars in the locker room.
"I'm in the middle of this, and it's a little awkward," Ross said. "The camera's in my pants. I have to get it out of my pants, and he's looking at me like, 'What are you doing?' And I'm like, 'You know what, Jim? This is my job, and I'm going to do it the best I can.' "
Ross, who has worked as a commentator for WWE since the mid-1980s, added, "They're two of the nicest guys I've ever been around in my life, and that's the way it should be. I mean, you can't get more professional than that." |
| **Typical** | The match ended in tears when The Deadman dropped the WWE Champion with a Tombstone Piledriver. Lesnar and The Undertaker did not work together again until 2005. The studio also screened a segment from Brakkton's show about Lesnar's first match with the Ultimate Warrior at In Your House 12: A Reunion.
Speak Therm
Good evening, State of the State. I'm Rick S. Ryals, Philadelphia's top city blogger. Now if you know anything about central Pennsylvania, you know that it's all about the chocolate chip cookies we call DEKERS. And to the prince himself, Duane Zane Dickenson, Philadelphia. This guy's the inspiration behind Strawberry Shortcake. So you know "Why have a Philly Man?" The Answer: Prince wasn't born or raised in the City of Brotherly Love. So all he knows about the area is from that TV show.
Whenever the topic of location comes |
| **Nucleus** | The match ended in tears when The Deadman dropped the WWE Champion with a Tombstone Piledriver. Lesnar and The Undertaker did not work together again until 2005. The lingering animosity led to a January 2007 match which ultimately concluded in Lesnar's first loss with the company. In November 2011, Lesnar wrestled The Undertaker at Wrestlemania XXVIII. After the end of his successful run in WWE, Lesnar was stripped of the World Heavyweight Championship due to an in-ring injury suffered in a match with Paul Heyman on April 18, 2008. When he regained the title, he returned to the ring in Extreme Rules against eventual WWE Champion CM Punk in a Hell in a Cell Match. This is Lesnar's only WWE Championship match.
Watch: Lesnar vs. Punk: Hell in a Cell 2011
Earlier this year, Brock Lesnar announced his retirement from MMA following a TKO loss to the former Strikeforce champion |
| **DAEMON** | Originally, the match would have be called "Opinion Day" and the Undertaker would have cashed in his infamous Money in the Bank contract to beat Lesnar. After all, that was what he born the Roman Reigns, beating him to the top in the main event of the November 14, 2007 pay-per-view. But then Lesnar would again have his match been taken on an unrelated alleged injury, kinder than Bill Goldberg's foot, and would use it as a springboard to the WWF Championship. Injuries would also again become a factor in the decision regarding Lesnar's match at Hell in a Cell, too, with Vince McMahon CFO George Barrios stating that Lesnar had "too much baggage" to compete with the Undertaker and Triple H. Ultimately, Lesnar refused to face Triple H, and under pressure from his stock market holdings (in spite of having the WWE Title on the line on SmackDown), cashed |

Table 12: Randomly selected generations from the test set of the Wikipedia domain.

| | |
|---|---|
| *Prefix* | Attalea species have a long history of human utilisation. Carbonised Attalea maripa seeds have been found in archaeological sites in Colombia dating |
| **CD** | to the Spanish/Portuguese settlements of the New World, and to the stone age. Today's strictly cultivated Attalea diacetiana seeds are a member of the digestive plant family being of the genus Attalea.
Originating in the New World, Attalea species began to be cultivated in Europe between c. 250 AD and 1600 AD * before entering the middle east during the Ottoman Empire (16th/17th century). From the 18th century onward, farmers in North America, South America, Madagascar, Africa, and Northern Australia began to use Attalea seeds to increase their global food production. From Pakistan maps exist showing Attalea making their way into Gujarat in North India, and from the Southern Indian region ASE (India), Pakistan, Bangladesh, Sri Lanka, and China. True to their early, Middle Eastern roots, the Attalea family has spread around the globe, possesing a wide diversity of beautiful flowers |
| **CS** | back to 3000 B.C. (1, 2).
It is thought that the use of Attalea seeds as a food source was introduced to New Zealand by Polynesians who settled the South Island in the 17th century (3). In recent years there has been a resurgence of interest in the plant, with the growing number of research papers and the publication of a book by Dr John Beddingfield in 2008 (4). This book was the first to provide a comprehensive overview of the history of the plant and its use in New Zealand.
A recent study has shown that the plant is a good source of vitamin C, which is essential for a healthy immune system (5). This is in contrast to the widely held belief that it is a vitamin B-deficient food source and should be avoided by vegans, vegetarians and those with low intake of vitamin B-12 due to the risk of anaemia (6). |
| **Typical** | back to the Middle and Archaic periods, when cranial and regal remains may have been deliberately buried. The seed also has a history of use as a powder by leather tanners.
Now, a team led by Dr John Choi, a curator at The National Museum, Singapore, has investigated the extraction and risk of bee venom in Attalea seeds.
'Early on, the bee's venom would have been used before treating wounds in leather engineering,' says Dr Choi, who presented his research at the 101st Scientific Sessions of the American Chemical Society 2017. 'After the invention of the stronger synthetic bee venom in the 19th century, researchers for some time were unconcerned about the risk of allergic reactions.'
Dr Lee Chin-yen; The National Museum
Dr Lee Chin-yen, who led the research, says: 'Because there is bias in the content of the Chinese literature on Apis cerana, most sensitive |
| **Nucleus** | back to the Middle and Archaic periods, when cranial and dental artefacts may have helped people to start cooking the plant.
There are many reasons why trees have been used for food or shelter, including their ability to grow quickly, their high water content and relatively high yield of crops, although forage plants are more typically grown for their nuts and seeds.
'Early people chewed on attalea to help them reach their food in past times,' says Zoran Kalerah, paleobiologist at the Natural History Museum of Denmark, who led the study.
'It was actually the starch in the leaves that aided their ability to reach their food. Basically what they do is they pulverise the attalea leaves and chew them to produce starch.
'This is the same stuff that they use for making baked goods and pastries, as the sticky content of the leaves allows them to drip onto a flat surface when |
| **DAEMON** | to the Spanish/Portuguese settlements of the New World, and to the stone age. Today's strictly cultivated Attalea diacetiana seeds are a member of the digestive plant family being of the genus Attalea.
Originating in the New World, Attalea species began to be cultivated in Europe between c. 250 AD and 1600 AD * before entering the middle east during the Ottoman Empire (16th/17th century). From the 18th century onward, farmers in North America, South America, Madagascar, Africa, and Northern Australia began to use Attalea seeds to increase their global food production. From Pakistan maps exist showing Attalea making their way into Gujarat in North India, and from the Southern Indian region ASE (India), Pakistan, Bangladesh, Sri Lanka, and China. True to their early, Middle Eastern roots, the Attalea family has spread around the globe, possesing a wide diversity of beautiful flowers |

Table 13: Randomly selected generations from the test set of the Wikipedia domain.

| | |
|---|---|
| *Prefix* | Amos was born in Macclesfield, Cheshire and was a member of the Crewe Alexandra academy until he was released at the age of 10. While |
| **CD** | at Crewe, Amos played for the Macclesfield Town youth teams. After leaving the academy he signed for the Crewe Alexandra reserves and was then released by them. Amos joined the Manchester City reserves and was then released. Amos then signed with Sheffield Wednesday, but was released at the age of 16 and joined the Crewe Alexandra academy again. After a spell with the academy Amos was released again, and then signed by Manchester City for the season in which they lost the League Championship final. Amos was released by City again and joined Sheffield United. He was released from United after two months and signed by Sheffield Wednesday, who he stayed at until he left them at the age of 23, having made over 200 league appearances. Amos has also represented the England youth sides at under 16, 17, 19, 21, 23, 25 and 28, and the under 19s at under 19, 21 and 25, and has been capped at every age group except for under |
| **CS** | his career was on the up, a knee injury in his first-team training led to him being loaned to Barnsley, where he made his debut for the Tykes in a 3-0 win over Rochdale on the opening day of the 1992-93 season.
The following season, Barnsley went on to win the League Two title and Amos was named PFA Young Player of the Year after netting 15 goals in all competitions. His goalscoring exploits earned him a move to Sheffield Wednesday for the 1993-94 season, where he played a key role in the club's promotion to the Premier League.
In the summer of 1994, Amos signed for West Ham United on a free transfer after his contract with Barnsley ran out. He went on to become the Hammers' top scorer in the Premier League that season, netting 14 goals as the Hammers finished runners-up to Manchester United. |
| **Typical** | a youngster (age 6) he appeared in this video on YOG TV. His first professional appearance came after coming off the bench to score a hat-trick for the Peterborough United youth team. He then appeared for Heracles Antwerp in the Belgian Top League and then went on trial with Celtic before the 1974 season. As a pyrotechnician he formed part of Charles and David White's all-time FA Cup XI. He then went on to play for Wrexham, Shrewsbury Town and at Bradford City after he was released from Heracles Town at the end of the 1974 season.
As with FM, Milton Keynes Dons have had their share of fireworks in the FA Cup. They have notched up 89 playoff goals, which is eight more than this year's champions
HYPOTHESIS: Tottenham will not make the final
HYPOTHESIS: United will be in the final |
| **Nucleus** | a youngster, he spent six months on the boats travelling on the Severn and Tyne rivers.
Spends the majority of his time in Ashton-under-Lyne. He works for Alfreton Community charity as part of the Lorna Beddoe project.
Internationals appearances: 16
Managerial past
Prior to moving into coaching, Amos worked as a referee and with the FA's student programme. He worked with the SFA's student team on a mentoring programme for under-15s and he was the Assistant Referee for the Under-16s and Under-18s. He was also part of the Professional Game Match Officials (PGMO) program, which is for those who wish to progress in an official career but have not yet signed professional contracts.
What he does
Amos has previously managed the reserves for Cheshire in the North West Counties League Premier Division and was |
| **DAEMON** | growing up, Amos worked as a bookmaker in a casino in Macclesfield.
His first professional football coach was John Olivieri ( playing for Crazy Gang, Liverpool and AC Milan) working under Fingleton and Hart for the Crewe Rams.
Amos moved to Accrington Stanley and made his professional debut as a substitute in March 1971. He made his debut in a 3-0 home win over Holland Farm in July that year.
In the summer of 1972 Amos signed for White Hart Lane. Amos achieved success in the League with the team, scoring 7 goals in 14 appearances in 1971-72. Amos' first team career was over before it started however for him.
After Martin Atkinson resigned in 1974, Amos was promoted to the Championship as a utility player. In his second season of the promotion he scored 10 goals in 20 games against three of England's Premier clubs; Hereford, Manchester United and Leeds United. |

Table 14: Randomly selected generations from the test set of the Wikipedia domain.

| | |
|---|---|
| *Prefix* | Oxaziridine derivatives were first reported in the mid - 1950s by Emmons and subsequently by Krimm and Horner and Jur |
| **CD** | gen Schulten. Oxaziridine derivatives were first reported in the mid - 1950s byand subsequentbyandandwere used as antifertility compounds in the mid-to late-1970's and 1980s, respectively, for treatment of male factor infertility. The compounds were later shown to have a variety of other biological activities. In vitro, the compounds inhibited cell cycle progression in various human and rat cell lines, inhibited cell growth in various cell lines in culture and inhibited proliferation and invasion in various animal cell types, including breast cancer and colon carcinomas. In addition to these in vivo activities in animal cell cultures and animal tumor models, in vitro data from several in vitro studies indicated the compounds to possess estrogen-like activity, including inhibition of human estrogen receptor a (HERS)-stimulated cell proliferation in a concentration dependent fashion. The in vitro estrogenicity was not due to direct binding to ERs but was mediated via a non-ER binding mechanism |
| **CS** | gens.
Aminoglycosides and aminoglycoside anhydrides have been used as anticancer agents for more than 50 yeard years. The use of these compounds has been based on their ability to inhibit growth of cancer cells and/or induce apoptosis (programmed cell death), which is a mechanism by which cancer cells escape from the immune system and proliferate in the absence of chemotherapeutic agents. However, there is a paucity of data in this area, which is the reason for the present review. This review will focus on the pharmacology/tumorigenicity of a number of compounds derived from the class of aminoalkyl esters, which are structurally related to benzimidazoles but differ in their structure and mechanism of action. Aminoglycosides and aminoglycoside anhydrides have been used as anticancer agents for more than 50 yeard years. The |
| **Typical** | gens. For many years, no complete pharmacological characterization of oxaziridine was capable. In 1982 Fabro et al. described the first idea ( Epstein & O'Brien, 1993 ) of a new mitochondrial impaired glutamate signalling pathway. The blue and green fluorescence specimens seen by Fabro et al (1983) are for the Fu-related alkaloid ( henbane and nocioin ). Disseminated from amphibila (The crabfoot vision constrictor) tadpoles, the xenon-and hydrogen peroxide-(2-hydroxyethanol) are produced from oxygen by electron acceptor monooxygenases (Bu (). Nitric oxide is transported by peroxynitrite.- Sudwarasingh Zingerke, Brian E. Jackson 1, Gerhard Rittner 1 Department of Pharmacy, University of Georgia School of Pharmacy, Athens, Georgia 30602-7155, www.uga. |
| **Nucleus** | gens. For many years, no complete pharmacological characterization of oxaziridine was reported. In 1982, a 1 1/2 hour infusion of oxaziridine hydrochloride ($5 \times 10\,9$ -$10\,9$ g/100 mL) in human volunteers was found to reduce excretion of dietary (glucose) and added-fat (citric acid) sugars and preserved 26% of exercise time. Several accidental ingestion of oxaziridine (1 g) by haemodialysis patients resulted in blood-poisoning and death. 17 In 1991, doses of 10 to 100 mg/kg in haemodialysis patients were found to cause cardiotoxicity in 2 cases. 18 In 2004, Bico et al. reported a 2 -year post-mortem study of the muscles of oxaziridine abusers showing heart enlargement with increased pulmonary artery "blaming" numbers and a lack of complex V-fibon |
| **DAEMON** | gens. Neither reported full anisidine binding to the terminal region of kappa-opioid receptors. Instead, oxaziridine was one of two anisole derivatives that failed to bind to rat brain A-alpha(1)DA receptors under binding conditions similar to those used for benzodiazepine binding in high concentrations in rat brain (Hoffman, 1996). Yet, oxaziridine did bind to human receptors of the A-beta(1)D OH and A-beta(1) isoforms, and the anisoles displayed similar pharmacological efficacy (Krimm and Horner, 1954). Oxaziridine and other anisole derivatives were subsequently shown to bind to rat brain and human brain A-beta(1) D OH receptors with similar efficacy. However, oxaziridine and other anisole derivatives were less efficient K(4) receptors and inhibited competitive binding to human [3H]dom |

Table 15: Randomly selected generations from the test set of the Wikipedia domain.

