# OpenReview forum: "Language Model Decoding as Direct Metrics Optimization"
_ICLR.cc/2024/Conference — ICLR 2024 poster_

### Official Review · Reviewer_aJdn · 2023-10-22

**Soundness:** 4 excellent
**Presentation:** 4 excellent
**Contribution:** 3 good
**Rating:** 8
**Confidence:** 3

**Summary:**

This paper proposes to frame the decoding for natural language generation as a direct metric optimization problem, aiming to generate texts that can align with human texts. To achieves this, the authors select a set of evaluation metric that reflects the quality of generation as constraints and then find the optimal solution to the optimization problem. The authors evaluate their method on a bunch of tasks and report performance in several metrics. According to the results, the proposed method achieves better performance than other decoding strategies such as Greedy, Top-k, and Nucleus decoding.

**Strengths:**

- The paper is well-written and easy to follow.
- The proposed method seems to be intuitive and effective.
- The authors provide detailed proof of all the propositions, which is highly appreciated.

**Weaknesses:**

- I don't have any big concerns about the paper. However, I have a question regarding the choice of the metric (see questions below).

**Questions:**

- Some evaluation metrics might contradict each other. For example, diversity and coherence are hard to achieve at the same time, as far as I know. I am wondering how the results will be if a different set of metrics is used.

---

> ### Author Response · Authors · 2023-11-15
> **Response to Reviewer aJdn**
>
> We thank the reviewer for the appreciation of our work!
>
> Regarding the question raised by the reviewer:
>
> >Q1. Some evaluation metrics might contradict each other. For example, diversity and coherence are hard to achieve at the same time, as far as I know. I am wondering how the results will be if a different set of metrics is used.
>
> 1. We should clarify that our goal is **not** to **maximize** all these metrics simultaneously. Instead, our aim is to realize **alignment** with human texts, i.e., our method should generate texts that achieve the same level as human texts evaluated under different metrics. Compared with maximization, *the goal of alignment is achievable in principle*, as human texts themselves perform as a natural demonstration. And therefore, this principle is independent from what types of metrics are employed in our method.
>
> 2. Coherence and diversity do not necessarily contradict with each other, but are two important aspects in evaluating text generation models [1]. Merely optimizing towards one metric will result in highly biased models [2]. Due to the limitation of model capacity and expressiveness, it is challenging for existing models to maximize both coherence and diversity, so that it might leave us the impression that these two metrics contradict each other. However, the goal of aligning to the coherence and diversity of human texts is achievable (mentioned by statement 1). As demonstrated in our empirical evaluations, our method is able to achieve **human-level** coherence and diversity (Figure 2 of our original manuscript).
>
> **References:**
>
> [1] Montahaei et al., Jointly Measuring Diversity and Quality in Text Generation Models. NAACL 2019.
>
> [2] Caccia et al., Language gans falling short. ICLR 2020.

---

> > ### Comment · Reviewer_aJdn · 2023-11-20
> >
> > Dear authors,
> >
> > Thank you for your response. My previous questions have been answered.
> >
> > I have a new question though regarding the added experiments in Appendix H. When one only optimizes for a single metric at decoding time, it seems the performance difference is small. One would expect that the score of, e.g., COH is higher when you optimize COH metric. But it seems it is not the case. Do you have any explanations or intuitions for this?

---

> ### Author Response · Authors · 2023-11-20
> **Response to Reviewer aJdn**
>
> We highly appreciate the reviewer’s timely response and the new question for us to clarify the purpose and design of our newly reported experiments!
>
> As noted in the first point in our previous response: our optimization goal is **not** to **maximize** all these metrics, but instead **align** with the performance of human texts on these metrics, simultaneously. As a result, when the COH metric is optimized, it is supposed to be the closest to the COH of the human reference, which is confirmed in our experiment. Please feel free to let us know if this helps resolve your concern!

---

> > ### Comment · Reviewer_aJdn · 2023-11-20
> >
> > Thanks for your reply. I think it is good work and I will keep my scores the same (8: accept, good paper).

---

### Official Review · Reviewer_iYh2 · 2023-10-31

**Soundness:** 3 good
**Presentation:** 3 good
**Contribution:** 2 fair
**Rating:** 5
**Confidence:** 3

**Summary:**

The paper proposes a decoding framework for language models that frames it as an optimization problem and performs decoding by optimizing towards desired dimensions. They prove that their method can improve the perplexity on human texts and also experimentally demonstrate that their method is effective on open-ended text generation tasks according to metrics such as repetition, coherence, and diversity.

**Strengths:**

1. The proposed framework makes sense and is technically sound. Based on their constructed framework, they propose a reasonable approximation that can work in practice.
2. Experiments demonstrate good empirical results compared to many other decoding algorithms.
3. Both automatic and human evaluations are conducted to provide insights into their method.

**Weaknesses:**

1. Related works such as [1] are missing. It is worth discussing and comparing with these methods in the paper.
2. The paper only conducts experiments in two open-ended text generation tasks, where both automatic and human evaluations are hard. On the other hand, evaluations on other text generation tasks such as machine translation and text summarization are more accurate and can better indicate if their method is indeed effective.
3. Their method can be more computational than baselines.




[1] Fernandes et al., Quality-Aware Decoding for Neural Machine Translation, NAACL 2022.

**Questions:**

How efficient is your algorithm?

---

> ### Author Response · Authors · 2023-11-15
> **Response to Reviewer iYh2 (part 1)**
>
> We are grateful to the reviewer for their feedback and suggestions, which will help improve our work.
>
> Regarding the weakness and questions raised by the reviewer:
>
> > W1. Related works such as [1] are missing. It is worth discussing and comparing with these methods in the paper.
>
> We appreciate the reviewer for pointing out this related work. And we have provided our discussion about it in the latest version of our submission in Appendix B.3.
>
> Although the work suggested by the reviewer used evaluation metrics to improve generation, we have distinct differences from the following perspectives:
>
> 1. On the theoretical guarantees:
>
> The reranking approach proposed in the suggested paper cannot guarantee improving the distribution of the base generation model to resemble the underlying text distribution from humans. Their objective was to optimize the final outputs towards some pre-defined metrics, but unfortunately they did not provide further theoretical analysis of the resulting performance.
>
> While our method has the theoretical guarantee to obtain a better approximation to the text distribution from humans than the base language model. This is presented in Section 2.2 where we proved that the optimal decoding distribution has a lower perplexity on human texts than the perplexity of the base model distribution. In Section 3.5, we also empirically validated this perplexity improvement.
>
> 2. On the optimization objectives:
>
> The reranking approach proposed in the suggested paper focused on improving generation quality, where they tune the weights of different metrics to maximize a given reference-based metric.
>
> While we consider alignment with human texts in various aspects simultaneously, including but not limited to the quality of text. We believe that alignment with a wide range of aspects, such as coherence, diversity, and repetition, is necessary to achieve **human-like generation**.
>
> The weights of different metrics are determined by exactly *matching the level of aspects of human texts*, instead of simply maximizing any particular metric. Therefore, we do not rely on any pre-defined reference-based metrics, as golden references with high quality and desired coverage are not always available in general text generation tasks.
>
> > W2. The paper only conducts experiments in two open-ended text generation tasks, where both automatic and human evaluations are hard. On the other hand, evaluations on other text generation tasks such as machine translation and text summarization are more accurate and can better indicate if their method is indeed effective.
>
> We follow recent works on decoding methods [1,2] to adopt open-ended generation tasks in our experiment given that they encompass a broader range of linguistic complexity and diversity. These tasks also set challenges for current modern language models to align with humans with respect to a wide range of aspects. Moreover, the open-ended setting is more conformed with the trending and natural use case of large language models which is less constrained and more flexible compared with closed-ended generation tasks such as text summarization.
>
> Nevertheless, to demonstrate the universability of our method, we conducted a new experiment on the standard summarization benchmark CNN/DailyMail using an off-the-shelf summarization model "pegasus-cnn_dailymail" from the [official release](https://huggingface.co/google/pegasus-cnn_dailymail) in Appendix I.
>
> Compared with the baseline methods, our method achieves better performance in alignment with various aspects like repetition, coherence, diversity, and information content (better when the score is closer to that of the reference), and ROUGE scores (better when the score is higher).
>
> | Model | seq-rep-3 | tok-rep-8 | coherence | diversity | $e^{ENT}$ | ROUGE-1  | ROUGE-2 | ROUGE-L |
> | --- | --- | --- | --- | --- | --- | --- | --- | --- |
> | Reference | 0.10 | 2.93 | 80.1 | 99.1 | 31.2 | N/A | N/A | N/A |
> | Top-k | 0.15 | 3.10 | 79.6 | 99.2 | 35.0 | 38.7 | 15.0 | 35.6 |
> | Nucleus | 0.15 | 3.00 | 72.3 | 99.0 | 104.3 | 34.4 | 12.2 | 31.6 |
> | Typical | 0.11 | 2.71 | 76.3 | 99.2 | 81.8 | 34.3 | 12.1 | 31.5 |
> | Contrastive Search | 0.15 | 3.14 | 80.9 | 98.8 | 28.4 | 40.9 | 16.9 | 37.8 |
> | Daemon (Ours) | **0.10** | **2.98** | **80.1** | **99.2** | **29.7** | **41.9** | **18.4** | **38.9** |
>
> (continue in part 2)

---

> > ### Author Response · Authors · 2023-11-15
> > **Response to Reviewer iYh2 (part 2)**
> >
> > > W3 & Q1. Their method can be more computational than baselines. How efficient is your algorithm?
> >
> > In our original submission, we have reported the decoding latency of our method with different $M$ in Figure 3 (the right-most subfigure). The inference latency is relative to greedy decoding. In our main experiments, we set $M=25$ and the latency of Algorithm 2 was only 1.35 times of the latency of greedy decoding, while our method achieved significantly better performance in overall metric alignment than greedy decoding and outperformed greedy decoding in Mauve score by nearly 30 points.
> >
> > To further compare with different decoding baselines in terms of the complexity and practical runtime, we updated an analysis in Appendix F of our updated manuscript. The decoding latency of our method is still on par with the advanced baselines, e.g., the latency of Contrastive Decoding [1] and Contrastive Search [2] are nearly 1.3 times of the latency of greedy decoding.
> >
> > **References:**
> >
> > [1] Li et al., Contrastive Decoding: Open-ended Text Generation as Optimization. ACL 2023.
> >
> > [2] Su et al., A Contrastive Framework for Neural Text Generation. NeurIPS 2022.

---

> > > ### Comment · Reviewer_iYh2 · 2023-11-20
> > >
> > > Thank you for your response!
> > >
> > > Could you clarify:
> > >
> > > 1. What's preventing Fernandes et al's method from using metrics such as repetition, coherence, diversity, and information content to improve their generation?
> > >
> > > 2. Why are your reported ROUGE scores for pegasus-cnn_dailymail significantly worse than what they reported in the paper (https://proceedings.mlr.press/v119/zhang20ae/zhang20ae.pdf) and on hugginface (https://huggingface.co/google/pegasus-cnn_dailymail)?

---

> > > > ### Author Response · Authors · 2023-11-21
> > > > **Further Response to Reviewer iYh2**
> > > >
> > > > We appreciate the reviewer’s follow-up comments and please find our answers to each of the questions below:
> > > > 1. As explained in our previous response, Fernandes et al's method only focused on maximizing a given reference-based metric. This focus limits its capacity to simultaneously improve various other aspects of text generation. In other words, its goal is not to align text generation performance across multiple aspects with human texts. Concretely, *their method treats those various metrics as features to improve the chosen reference-based metric* (from the paragraph 2 in Section 2.2.1 of Fernandes et al). As the reference-based metric only emphasizes text matching against the reference which overlooks other critical dimensions, such as coherence, diversity, etc., the generation performance measured under those metrics cannot be guaranteed to match with that on human texts.
> > > > 2. The focus of our study concentrates on language model decoding. Hence, we compared our method against established decoding methods for language models, such as top-k sampling [1], Nucleus sampling [2], and Contrastive Search[3], etc in general text generation tasks. The suggested Pegasus work [4] is not about language model decoding, but specifically text summarization. In their solution, they employed beam search, which has been criticized for ignoring the rich information in the distribution of language model and often leads to degenerative repetition in the literature of language model decoding [2]. Despite this, we adhered to the implementation described in [4] and evaluated beam search on the metrics we considered in our experiments. From the result, we can clearly observe that beam search is far away from the metric scores of the reference texts in terms of repetition, coherence, diversity and information content, and has the **worst performance** in aligning with human texts on all these aspects. While it did achieve high ROUGE scores, the aforementioned aspects cannot be covered by the ROUGE metric that only concerns local text matching against the reference. In fact, the ROUGE metric has been criticized for overly rewarding lexical similarity [5] while failing to adequately capture aspects like diversity and semantic coherence [6]. Though our objective is not to maximize ROUGE scores, it's noteworthy that, comparing to our baseline decoding methods, our method still achieved the best performance in ROUGE evaluations by targeting alignment across diverse text generation aspects.
> > > >
> > > > | Model | seq-rep-3 | tok-rep-8 | coherence | diversity | $e^{ENT}$ | ROUGE-1  | ROUGE-2 | ROUGE-L |
> > > > | --- | --- | --- | --- | --- | --- | --- | --- | --- |
> > > > | Reference | 0.10 | 2.93 | 80.1 | 99.1 | 31.2 | N/A | N/A | N/A |
> > > > | Top-k | 0.15 | 3.10 | 79.6 | 99.2 | 35.0 | 38.7 | 15.0 | 35.6 |
> > > > | Nucleus | 0.15 | 3.00 | 72.3 | 99.0 | 104.3 | 34.4 | 12.2 | 31.6 |
> > > > | Typical | 0.11 | 2.71 | 76.3 | 99.2 | 81.8 | 34.3 | 12.1 | 31.5 |
> > > > | Contrastive Search | 0.15 | 3.14 | 80.9 | 98.8 | 28.4 | 40.9 | 16.9 | 37.8 |
> > > > | Daemon (Ours) | **0.10** | **2.98** | **80.1** | **99.2** | **29.7** | **41.9*** | **18.4*** | **38.9*** |
> > > > | Beam search (used in [4]) | 0.26 | 3.27 | 81.4 | 98.6 | 25.5 | **43.4** | **20.1** | **40.4** |
> > > > *: the best performance comparing to our baseline decoding methods
> > > >
> > > >
> > > > **References:**
> > > >
> > > > [1] Fan et al., Hierarchical Neural Story Generation
> > > >
> > > > [2] Holtzman et al., The Curious Case of Neural Text Degeneration
> > > >
> > > > [3] Su et al., A Contrastive Framework for Neural Text Generation
> > > >
> > > > [4] Zhang et al., PEGASUS: Pre-training with Extracted Gap-sentences for Abstractive Summarization
> > > >
> > > > [5] Ng et al., Better Summarization Evaluation with Word Embeddings for ROUGE
> > > >
> > > > [6] Zhang et al., ROUGE-SEM: Better evaluation of summarization using ROUGE combined with semantics

---

> > > > > ### Comment · Reviewer_iYh2 · 2023-11-21
> > > > >
> > > > > Thanks for the response! I'd like to expand my previous comments just in case they are not clear:
> > > > >
> > > > > 1. My point is that Fernandes et al's method is a general framework that can take reference-based and reference-free metrics (which they both experimented with), including the metrics you used in the paper (repetition, coherence, etc.). Therefore, I do not think there is a difference in this regard and a better way of differentiating your work from theirs is required.
> > > > >
> > > > >
> > > > > 2. Beam search is a popular decoding method and is the standard decoding method in text summarization. I don't think you can exclude beam search from "language model decoding."
> > > > >
> > > > > ROUGE score, despite being imperfect, is still widely adopted and is one of the most important metrics in the summarization community that correlates with human evaluations.
> > > > >
> > > > > My intention of testing your models on tasks such as machine translation and text summarization was that evaluations in open-ended generation tasks are hard and unreliable, whereas BLEU in machine translation and ROUGE in summarization are relatively reliable metrics for reflecting your generation quality. Therefore, the provided results on summarization does not address my concern that your method can be ineffective in this setting.
> > > > >
> > > > > 3. A side note: ``human-like generation'' is a vague term, and I don't think your definition of this term can be applied to text summarization, especially considering previous works (e.g. [1]) have conducted human evaluations and have results contradicting your findings here.
> > > > >
> > > > >
> > > > > [1] Wiher, Gian, Clara Meister, and Ryan Cotterell. "On decoding strategies for neural text generators." TACL 2022.

---

> > > > > > ### Author Response · Authors · 2023-11-23
> > > > > > **Further Response to Reviewer iYh2**
> > > > > >
> > > > > > Thanks for the reviewer’s timely feedback! Please find our answers to the questions below:
> > > > > > 1. The method proposed by Fernandes et al. only facilitates **optimizing a single metric**, while our method targets at simultaneously **optimizing multiple metrics**, i.e., aligning the generated texts with human texts in terms of those metrics. Although their method is able to take multiple metrics as input, their objective is to only optimize a single chosen metric (Equation 3 of Fernandes et al.). As described in their method section (Section 3 of Fernandes et al), "The weights in Eq. 3 are optimized to maximize a given reference-based metric $M_{ref}$". Also, in the experiments (Table 1, group 2 of Fernandes et al.), T-RR w/ BLEU, T-RR w/ BLEURT, T-RR w/ COMET are optimized towards BLEU, BLEURT and COMET respectively. In their reported results, we can clearly notice that none of those combinations could simutaneously achieve strong performance in the overall assessment across all metrics, and their performance exceled only in the specific metric which was optimized for. On the contrary, our method optimizes towards the alignment of these metrics against human texts simultaneously (Equation 1 of our paper). Specifically, all metrics constrain the optimization problem defined in Equation 1, so that they are all optimized. This has been verified in our experiments (e.g., Table 1 of our paper): our method achieves the best overall performance in aligning with human texts on all aspects measured by the corresponding metrics.
> > > > > >
> > > > > > 2. We acknowledge that beam search is a standard choice in summarization, but we should also note that the primary focus of our study is on decoding from pre-trained language models in general. On the specific task of summarization, given the strong performance of beam search on ROUGE scores, we also extend Algorithm 2 of our method to adopt beam search to generate better candidates. From the newly obtained results reported in the table below, we observe that Daemon achieved **better** performance in aligning with human texts on all aspects (scores closer to that of the references), while maintaining **competitive** performance on ROUGE scores with beam search. Moreover, maximization-based decoding methods, like beam search, tend to produce degenerated repetitions (see Figure 1 in [2]). But as shown in our results, Daemon clearly allevates this issue (indicated by lower seq-rep-3 and tok-rep-8), thanks to its ability in aligning multiple aspects (including repetition) simultaneously.
> > > > > >
> > > > > > | Model | seq-rep-3 | tok-rep-8 | coherence | diversity | $e^{ENT}$ | ROUGE-1  | ROUGE-2 | ROUGE-L |
> > > > > > | --- | --- | --- | --- | --- | --- | --- | --- | --- |
> > > > > > | Reference | 0.10 | 2.93 | 80.1 | 99.1 | 31.2 | N/A | N/A | N/A |
> > > > > > | Beam search | 0.26 | 3.27 | 81.4 | 98.6 | 25.5 | **43.4** | 20.1 | **40.4** |
> > > > > > | Daemon (w/ Beam-search-based candidate generation) | **0.17**  | **3.10**  | **80.9**  | **99.0**  | **27.2**  | 43.2 | **20.8** | 40.2 |
> > > > > >
> > > > > > 3. The definition and evaluation of the alignment with human texts are established by [2] where the authors argue that *the optimal generation strategy should produce a distribution that is **close** to the human distribution in terms of various metrics*. We also followed their experiment settings to assess the closeness between the scores of generated texts and those of human texts on various metrics. This criterion has also been well-acknowledged [3,4] and applied to language model decoding on various generation tasks including summarization in the follow-up studies like [3]. Finally, we note that the human evaluation results on summarization in [1] do not necessarily contradict our alignment evaluation. The human evaluation in [1] mainly considers accuracy and quality (Figure 2 in [1]), but pays less attention to other aspects like diversity, information content, etc., which are covered in our alignment evaluation. In our main experiments (Section 3.4), we have conducted thorough human evaluations to confirm that the alignment evaluation on these aspects is consistent with human preferences.
> > > > > >
> > > > > > **References:**
> > > > > >
> > > > > > [1] Meister et al., On decoding strategies for neural text generators. TACL 2022.
> > > > > >
> > > > > > [2] Holtzman et al., The Curious Case of Neural Text Degeneration. ICLR 2020.
> > > > > >
> > > > > > [3] Meister et al., Locally Typical Sampling. TACL 2023.
> > > > > >
> > > > > > [4] Su et al., A Contrastive Framework for Neural Text Generation. Neurips 2022.

---

### Official Review · Reviewer_3A4S · 2023-11-02

**Soundness:** 3 good
**Presentation:** 3 good
**Contribution:** 3 good
**Rating:** 6
**Confidence:** 4

**Summary:**

This paper introduces a parametric decoding method for generating text from probabilistic models. Essentially, samples from a model are reweighted according to the degree to which they satisfy particular constraints, where the importance of these different constraints is learned via numerical optimization using a small set of samples. These constraints are that the texts generated by the model match human-generated texts according to a variety of chosen metrics. In order to sample from the new (sequence-level) distribution, they make use of a sampling based approximation of the distribution over continuations (of a particular prefix). They show strong results in terms of empirical performance in comparison to standard baseline decoding methods.

**Strengths:**

* The method is novel and provides a nice lightweight alternative to fine-tuning methods. It may thus be widely accessible to practitioners without the ability to tune larger language models
* The empirical component of the paper is comprehensive, including nice ablation studies

**Weaknesses:**

* The mathematical motivations given by this paper are quite weakly supported/explained. In general, the language used by the authors with respect to this topic is confusing and informal. For example, they motivate their use of the reverse KL for choosing the parameters of q by stating “KL(q || pθ) restricts the decoding distribution q to deviate minimally from the LM distribution pθ”, but the same argument could be made for the forward variant of the divergence. Similar language is scattered across 2.1
* The method requires sampling in order to approximate the new parametric distribution (via the WIS algorithm). The runtime of this algorithm, and hence the additional computational complexity incurred by this decoding method, is not discussed. Further, there is an additional sampling + renormalization step that must happen during decoding (algorithm 2). The lack of discussion of these costs is especially pertinent given that the authors point to the tuning required by other decoding methods as one of their downsides “But unlike those previous decoding methods that require heavy manual hyper-parameter tuning for trade-off among different metrics”

**Questions:**

* The authors point out that they wish for the difference in the expected value of f under the model and under the data distribution to be 0. It’s unclear to me how this is incorporated into the learning of the different \mu. I see that the target expectation is used in the learning algorithm, but this is not very intuitive. Could some intuition be provided here?
* “Most existing decoding algorithms lead to deteriorated ε-perplexity, comparing to directly sampling from the input LM distribution.” It seems that [1] shows otherwise
* In 2.2, I don’t see why “the second result reveals the perplexity improvement over pθ due to the non-negativity of DKL(pθ,μ || pθ)”
* The setup for human evaluation is not well explained. What do the numbers in table 3 mean?
* I do not follow the last statement on page 5 “...so that it can be directly plugged into Eq. (2) to calculate perplexity”

[1] Meister et. al. 2023. On the Efficacy of Sampling Adapters.

---

> ### Author Response · Authors · 2023-11-15
> **Response to Reviewer 3A4S (part 1)**
>
> We appreciate the comprehensive feedback and helpful recommendations provided by the reviewer that will enhance the quality of our work.
>
> Regarding the weaknesses and questions raised by the reviewer:
>
> > W1. The mathematical motivations given by this paper are quite weakly supported/explained. For example, they motivate their use of the reverse KL for choosing the parameters of q by stating “KL(q || pθ) restricts the decoding distribution q to deviate minimally from the LM distribution pθ”, but the same argument could be made for the forward variant of the divergence.
>
> Our motivation for using reverse KL instead of forward KL stems from the demand of high-quality texts when  decoding from a language model. We should note that as KL divergency is not symmetric, *forward KL and reverse KL measure the differences between two distributions quite differently* (Section 1.6.1 in [1]). And we have explained this in Section 2.1 of our original submission. To make it clearer, we further elaborate the differences between these two types of KL divergency in details below and connect them with our motivation. We also added the discussions to Appendix C of our updated manuscript.
>
> Minimizing the reverse KL, $KL(q || p_\theta)$ is known to encourage **zero-forcing** behavior [2], as it forces $q(x)=0$ when $p_\theta(x)=0$. This restricts $q$ to be **mode-seeking** and only explores the modes of the target distribution $p_\theta$ which contains samples with high likelihood under the ground-truth model's distribution and thus ignores the outliers.
>
> Whereas, minimizing the forward KL, $KL(p_\theta || q)$ encourages **zero-avoiding** behavior, i.e., it avoids $q(x)=0$ when $p_\theta(x)>0$. This leads $q$ to be **mean-seeking** which spreads its support to cover all the non-zero density regions of $p_\theta$. As the distribution $p_\theta$ of language models is known to have unreliable long tails [3], the resulted $q$ can further overestimate those tails to produce low-quality samples [4].
>
> We also added an intuitive visualization to illustrate the different behaviors of reverse and forward KL in Figure 4 of Appendix C of our updated manuscript.
>
> Overall, comparing to the forward KL, minimizing the reverse KL leads to a more focused distribution and ignores the long tail in the given language model distribution, which is more suitable for our decoding scenario that demands generation quality.
>
> > W2. The method requires sampling in order to approximate the new parametric distribution (via the WIS algorithm). The runtime of this algorithm, and hence the additional computational complexity incurred by this decoding method, is not discussed. Further, there is an additional sampling + renormalization step that must happen during decoding (algorithm 2). The lack of discussion of these costs is especially pertinent given that the authors point to the tuning required by other decoding methods as one of their downsides “But unlike those previous decoding methods that require heavy manual hyper-parameter tuning for trade-off among different metrics”
>
> We have updated the runtime analysis of Algorithm 1 (the WIS algorithm) in Appendix E of our updated manuscript, so as to justify our solution's benefit over previous decoding methods that count on manual hyper-parameter tuning.
>
> First, we have noted in Section 2.3.1 of our original submission that Algorithm 1 is only used on the development set to determine the parameters $\\{\mu_i\\}^K_{i=1}$ defined in the energy function (**prior to** the decoding phase). Once settled, these parameters are fixed in the inference stage for decoding on the test set. As a result, Algorithm 1 **does not** incur additional computation at the decoding stage.
>
> Second, the computational cost of Algorithm 1 is much lower than the manual hyper-parameter tuning methods employed in most of our decoding baselines. Specifically, the computational cost in Algorithm 1 consists of two parts.
>
> - The first part takes **one run** of unconditional sampling from the base language model (line 2 of Algorithm 1). We typically set the number of samples to be the same as the development set, so that this cost is equal to one time sampling on the development set.
>
> - The second part estimates the parameters $\\{\mu_i\\}^K_{i=1}$ via iterative gradient descent (line 3-6 of Algorithm 1). As the number of parameters is small (equals to the number of constraints which is usually in the scale of dozens in practice), and the metric scores $\\{f_i(x)\\}^K_{i=1}$ can be precomputed. The total computational overhead here is very small. In our experiments, it takes less than 1 minute to perform thousands of gradient descent steps, which can be ignored comparing to the typical cost consuming when sampling from a language model.
>
> (continue in part 2)

---

> > ### Author Response · Authors · 2023-11-15
> > **Response to Reviewer 3A4S (part 2)**
> >
> > However, on the other hand, typical grid search of hyper-parameters requires **$H\times R$ runs** of sampling in the development set, where $H$ is the number of hyper-parameters and $R$ is the number of trials for each hyper-parameter. Even the labor intensive manual tuning methods require **at least one run** of sampling in the development set, which do not have any advantage over our Algorithm 1 in terms of computational cost.
> >
> > Third, Algorithm 2 is used to generate samples in the inference stage. We also updated the comparison of computational complexity and runtime analysis of our method and other baselines in Appendix F. To summarize, the main complexity of Algorithm 2 comes from the candidate sampling step (line 1-2 in Algorithm 2), which increases the computational complexity of vanilla greedy decoding by a factor of $M$ (number of candidates). However, as this step can be done in parallel, which is highly optimized by modern GPU hardware, the actual inference latency grows much slower with the increase of $M$.
> >
> > In the original submission, we have reported the latency of Algorithm 2 (sampling + resampling) with different $M$ in Figure 3 (the right-most subfigure). The inference latency is relative to greedy decoding. In our main experiments, we set $M=25$ and the latency of Algorithm 2 was only 1.35 times of that from greedy decoding, while our method achieved significantly better performance in overall metric alignment than greedy decoding and outperformed it in Mauve score by nearly 30 points.
> >
> > > Q1. The authors point out that they wish for the difference in the expected value of f under the model and under the data distribution to be 0. It’s unclear to me how this is incorporated into the learning of the different \mu. I see that the target expectation is used in the learning algorithm, but this is not very intuitive. Could some intuition be provided here?
> >
> > In essence, the parameters $\\{\mu_i\\}^k_{i=1}$ are learned to minimize the difference between the expected value of $\\{f_i\\}^k_{i=1}$ evaluated on the generated texts: $\\mathbb{E}\_{x\sim p_{\theta, \mu}}[f_i(x)]$ and the real data: $\mathbb{E}\_{x\sim p_d}[f_i(x)]$. As the decoding distribution $p_{\theta,\mu}$ is a function of $\mu$, the estimation error between the approximated expectation and target expectation can be back-propagated to update $\\{\mu_i\\}^k_{i=1}$ accordingly.
> >
> > > Q2. “Most existing decoding algorithms lead to deteriorated ε-perplexity, comparing to directly sampling from the input LM distribution.” It seems that [1] shows otherwise.
> >
> > First, our decoding method can be directly used to calculate perplexity (Proposition 2.1 establishes the feasibility of computing perplexity under the decoding distribution of our method), while existing truncation-based decoding algorithms have to use the surrogate ε-perplexity. Therefore, the issue of ε-perplexity does not affect any of our method's results.
> >
> > Second, our initial observation was drawn from the results presented in Table 2 of [5] where 13 out of 15 decoding methods have worse ε-perplexity than the model distribution. [5] also pointed out that the disadvantage of ε-perplexity is that "it still does not evaluate the original sparse distribution, but rather a modified version of it." So the value of ε-perplexity does not faithfully reflect the true perplexity of a truncated decoding distribution.
> >
> > In the paper suggested by the reviewer, Figure 1 shows that the **forward cross-entropy** of the unmodified model distribution is the **lower bound** of the **ε-smoothed forward cross-entropy** of different decoding methods with different configurations. Although presented differently, this result leads to the same conclusion drawn in [5], as *the forward cross-entropy is actually the log of perplexity*, and lower forward cross-entropy indicates better perplexity.
> >
> > Although the paper suggested by the reviewer also reported the performance of ε-variants on other distribution measures, the results in Figure 1 can not reach a clear conclusion on whether the ε-variant is better or worse on these distribution measures.
> > However, the discussion on these ε-variants is clearly out of the scope of this paper, and we choose to remove this sentence in the updated version of our paper to avoid misunderstanding.
> >
> > (continue in part 3)

---

> > > ### Author Response · Authors · 2023-11-15
> > > **Response to Reviewer 3A4S (part 3)**
> > >
> > > > Q3. In 2.2, I don’t see why “the second result reveals the perplexity improvement over pθ due to the non-negativity of DKL(pθ,μ || pθ)”.
> > >
> > > Perplexity is defined as $2^{H(p_d, q)}$ where $H(p_d, q)$ is the cross entropy of distribution $q$ evaluated on the data distribution $p_d$.
> > >
> > > The second result in our Proposition 2 states that the cross entropy $H(p_d, q_{opt})$ of our decoding distribution $q_{opt}$ (defined in equation 1) is **smaller** than the cross entropy $H(p_d, p_\theta)$ of the original language model distribution $p_\theta$, where the difference is $D_{KL}(q_{opt}\\|p_\theta)$. Note that $p_{\theta,\mu}$ equals to $q_{opt}$ when $\mu$ is optimal, and we have clarified the notation to $q_{opt}$ as well as the relation between perplexity and cross entropy in our updated manuscript to make it clearer in its context.
> > >
> > > > Q4. The setup for human evaluation is not well explained. What do the numbers in table 3 mean?
> > >
> > > We have described our human evaluation in Section 3.4 with detailed setup provided in Appendix C of our original submission (Appendix D of the updated manuscript), including the annotation process and the annotation guidelines in detail. To make it clearer, we explain the human evaluation setup again in the following part.
> > >
> > > Given the prefix randomly sampled from the test set, and two continuations generated by our method and one of the baselines, the human annotators were required to select the better continuation or indicate a draw, according to the evaluation criteria: coherence, fluency, and informativeness.
> > >
> > > The numbers in Table 3 are the average win / lose rate of our method against a chosen baseline method, which are calculated via dividing the total number of wins/loses by the total number of judgments.
> > >
> > > > Q5. I do not follow the last statement on page 5 “...so that it can be directly plugged into Eq. (2) to calculate perplexity”
> > >
> > > As modulating the model distribution with a temperature does not change its support, which preserves the feasibility to compute perplexity (Proposition 2.1 still holds). In the original submission, we have also included procedures to calculate the perplexity of the decoding distribution of our method in Appendix A.3, which explain the derivation in detail. We have added a reference to this sentence in the updated manuscript.
> > >
> > > **References:**
> > >
> > > [1] Bishop, Pattern Recognition and Machine Learning, 2016.
> > >
> > > [2] Malinin et al., Reverse kl-divergence training of prior networks: Improved uncertainty and adversarial robustness, NeurIPS 2019.
> > >
> > > [3] Holtzman et al., The Curious Case of Neural Text Degeneration, ICLR 2020.
> > >
> > > [4] Chan et al., Investigating Forward and Reverse KL Divergences, JMLR 2022.
> > >
> > > [5] Martins et al., Sparse Text Generation. EMNLP 2020.

---

> > > > ### Comment · Reviewer_3A4S · 2023-11-20
> > > > **Response**
> > > >
> > > > Thank you for the detailed responses. At the moment, my score remains the same

---

> > > > > ### Author Response · Authors · 2023-11-20
> > > > > **Further Response to Reviewer 3A4S**
> > > > >
> > > > > Thank you for your feedback. In our detailed response, we have thoroughly answered the reviewer’s main concerns regarding the theoretical justification (see answer to W1) for using reverse KL and provided an in-depth empirical runtime analysis of our algorithm (see answer to W2). These discussions are also reflected in our updated manuscript, which significantly strengthens our work. We aimed to address all questions raised by the reviewer, and we hope that our efforts have resolved any uncertainties or possible confusions of the reviewer.
> > > > >
> > > > > We would like to kindly ask the reviewer to confirm if our revisions and clarifications have fully addressed the reviewer’s concerns? If there are any further questions or aspects that might hinder a deeper appreciation of our work, we are more than willing to provide additional information or clarification in time.

---

### Official Review · Reviewer_MQ6q · 2023-11-12

**Soundness:** 3 good
**Presentation:** 3 good
**Contribution:** 3 good
**Rating:** 6
**Confidence:** 3

**Summary:**

The authors propose a decoding scheme (DAEMON) based on minimizing the divergence of the distribution over generated texts, $q$, with that of the base LM, $p$. They formulate as an optimization problem of this form with a KL objective, provide an approximation, and an efficient sampling scheme for use during decoding.

**Strengths:**

- The paper is well written and straightforward to follow.
- The motivation is sound, the authors provide an approach for estimating $\mu$, and for sampling 'efficiently' (although the time complexity of this approach is not given).
- Strong experimental results:
   - over a range of datasets
   - against quite a few decoding approaches commonly used in practice
   - ablation studies
   - a range of metrics (Repetition, Coherence, Diversity, Information Content)
- The problem is of great importance in the community.

**Weaknesses:**

- Please provide complexity analysis of the sampling approach and compare to competing approaches (Greedy, Top-k, CD, CS, etc). While the experimental results are strong it is important to compare the runtime of this method to determine practical efficacy given the authors claim it is "efficient" in the conclusion.
- Analysis on the convergence of $\mu$ in Algorithm 1 and some sensitivity analysis to initialization would be helpful for practitioners.
- Given that different metrics perform stronger for different # of candidates $M$ and $\tau$, it would be helpful, in terms of robustness, to include some experimental results which show performance on different metrics if one optimizes for a single metric (i.e. coherence).

**Questions:**

See weaknesses

---

> ### Author Response · Authors · 2023-11-15
> **Response to Reviewer MQ6q**
>
> We thank the reviewer for the detailed feedback and valuable suggestions to further improve our work!
>
> Regarding the weakness and questions raised by the reviewer:
>
> > Please provide complexity analysis of the sampling approach and compare to competing approaches (Greedy, Top-k, CD, CS, etc). While the experimental results are strong it is important to compare the runtime of this method to determine practical efficacy given the authors claim it is "efficient" in the conclusion.
>
> First of all, we have to clarify that we did **not** emphasize efficiency as an advantage of our method, although the decoding latency of our method is still on par with vanilla greedy decoding. In conclusion and future work discussions (Section 4), we only suggested finding more efficient sampling methods for our decoding framework as an important future work, rather than claiming we have already done so.
>
> Second, in our submitted manuscript, we reported the decoding latency of our method with different $M$ in Figure 3 (the right-most subfigure). The inference latency is relative to greedy decoding. In our main experiments, we set $M=25$ and the latency of Algorithm 2 was only 1.35 times of that from greedy decoding, while our method achieved significantly better performance in overall metric alignment than greedy decoding and outperformed it in Mauve score by nearly 30 points.
>
> We have updated the computational complexity and practical runtime analysis of different sampling approaches in Appendix F. To summarize, the main complexity of our method comes from the candidate sampling step (line 1-2 in Algorithm 2), which increases the computational complexity of vanilla greedy decoding by a factor of $M$ (the number of candidates). However, as this step can be done in parallel, which is highly optimized by modern GPU hardware, the actual inference latency grows much slower in terms of $M$. According to the right-most subfigure of Figure 3, the latency is still below 2 when $M$ is set to 100.
>
> > Analysis on the convergence of $\mu$ in Algorithm 1 and some sensitivity analysis to initialization would be helpful for practitioners.
>
> We have updated the analysis about the convergence and sensitivity of $\mu$ in Appendix G. We consider three common initializations of $\mu$:
> (1) All-zero initialization.
> (2) Random initialization from a standard normal distribution.
> (3) Random initialization from a uniform distribution $U[0,1)$.
>
> We took 5 runs with different random seeds in these three settings, and found that Algorithm 1 is quite stable across all these different initializations. To analyze the convergence of $\mu$, we plot the optimization trajectory of $\mu_i$ with different initializations in Figure 5 of Appendix G. The results suggest that the optimization of $\mu$ is quite stable under different initializations and they almost converge to the same optimal value, which demonstrates the practicality of Algorithm 1.
>
> > Given that different metrics perform stronger for different # of candidates $M$ and $\tau$, it would be helpful, in terms of robustness, to include some experimental results which show performance on different metrics if one optimizes for a single metric (i.e. coherence).
>
> We appreciate the reviewer's suggestion. Firstly, we note that our choice of $M$ and $\tau$ is already able to achieve strong performance in universal alignment on different metrics than existing baselines in the main results (Table 1).
>
> To demonstrate the robustness of the alignment results regarding to the choice of $M$ and $\tau$, we followed the reviewer's suggestion to include additional experiments in Appendix H of our updated manuscript, where we studied the performance of different metrics when one optimizes against a single metric. The results are presented in Table 5 of Appendix H. It shows that the performance on different metrics has a very small variance even when we optimize for a single metric, which indicates the robustness of our method.
>
> | Model | seq-rep-4 | tok-rep-32 | coherence | diversity | $e^{ENT}$ |
> | --- | --- | --- | --- | --- | --- |
> | Reference | 0.48  | 21.3  | 62.3  | 92.5  | 23.2 |
> | Opt. seq-rep-4 | 0.42  | 22.5  | 62.5  | 92.2  | 22.8 |
> | Opt. tok-rep-32 | 0.38  | 21.2  | 62.4  | 94.1  | 24.3 |
> | Opt. coherence | 0.38  | 21.2  | 62.4  | 94.1  | 24.3 |
> | Opt. diversity | 0.55  | 22.9  | 63.3  | 92.7  | 22.2 |
> | Opt. $e^{ENT}$ | 0.40  | 21.5  | 61.6  | 93.9  | 23.1 |

---

> > ### Author Response · Authors · 2023-11-22
> > **Further Response to Reviewer MQ6q**
> >
> > Dear Reviewer MQ6q,
> >
> > Thank you for the time and effort you've invested in reviewing our submission. We hope our previous responses have addressed your concerns. With the ICLR rebuttal deadline approaching, we eagerly anticipate your further thoughts on our work. Please feel free to let us know if there are any further aspects you'd like us to clarify.

---

### Author Response · Authors · 2023-11-15
**General response to all reviewers**

We sincerely thank all the reviewers for their thoughtful comments and constructive suggestions, which surely  help us strengthen our paper! It is encouraging that all reviewers appreciate the novelty and practical impact of our proposed decoding solution, extensive empirical evaluations, and encouraging performance. In addition, Reviewer MQ6q believed our studied problem is of great importance to the community and Reviewer aJdn particularly appreciate our detailed theoretical analysis. In the following, we will respond to each reviewer's comments and questions respectively. We have also updated our manuscript to incorporate the reviewers' suggestions. The updated content is highlighted in blue in our revised submission.

---

### Author Response · Authors · 2023-11-18
**Reminder Regarding the Review Discussion Timeline**

Dear reviewers,

We hope this message finds you well. We understand that the author-reviewer discussion is a critical component of the ICLR review process, and we would like to remind you that the rebuttal period is scheduled to conclude on **November 22nd**. Given there are only **4 days** left before the deadline, we would like to call for our reviewers' attention to our provided responses.

We are fully committed to engaging with the discussions, if any further information or clarification is needed regarding our response. Thank you for your time and attention to this matter! Your efforts in reviewing submissions are deeply appreciated.

---

### Meta-Review · Area_Chair_sFkT · 2023-12-15

**Metareview:**

The paper proposes a decoding framework for language models that frames it as an optimization problem and performs decoding by optimizing towards desired dimensions. They prove that their method can improve the perplexity of human texts and also experimentally demonstrate that their method is effective on open-ended text generation tasks according to metrics such as repetition, coherence, and diversity.
The paper is well written and straightforward to follow and the problem is of great importance to the community. The authors present strong experimental results over a range of datasets and a range of metrics including nice ablation studies. Some reviewers would like to see a deeper discussion of the runtime of the proposed algorithm. Some changes have already been made to the paper addressing this. A remaining open discussion is the comparison to re-ranking and Minimum Bayes Risk Decoding approaches [1,2] which in theory can also use multiple metrics during decoding. I would recommend putting these two papers and approaches (reranking and MBR decoding) better into perspective.

[1] https://aclanthology.org/2022.naacl-main.100/
[2] https://aclanthology.org/2022.tacl-1.47/

**Justification For Why Not Higher Score:**

The paper proposes aproaches where similar methods already exist (Reranking, MBR).

**Justification For Why Not Lower Score:**

There are no weaknesses in the paper and the experimental results are good and insightful.

---

### Decision · Program_Chairs · 2024-01-16

Accept (poster)